# A genetically encoded biosensor to monitor dynamic changes of c-di-GMP with high temporal resolution

Andreas Kaczmarczyk ◉[1] ✉, Simon van Vliet ◉[1], Roman Peter Jakob ◉[1], Raphael Dias Teixeira ◉[1], Inga Scheidat ◉[1], Alberto Reinders[1], Alexander Klotz[1], Timm Maier ◉[1] & Urs Jenal ◉[1] ✉

Monitoring changes of signaling molecules and metabolites with high temporal resolution is key to understanding dynamic biological systems. Here, we use directed evolution to develop a genetically encoded ratiometric biosensor for c-di-GMP, a ubiquitous bacterial second messenger regulating important biological processes like motility, surface attachment, virulence and persistence. The resulting biosensor, cdGreen2, faithfully tracks c-di-GMP in single cells and with high temporal resolution over extended imaging times, making it possible to resolve regulatory networks driving bimodal developmental programs in different bacterial model organisms. We further adopt cdGreen2 as a simple tool for in vitro studies, facilitating high-throughput screens for compounds interfering with c-di-GMP signaling and biofilm formation. The sensitivity and versatility of cdGreen2 could help reveal c-di-GMP dynamics in a broad range of microorganisms with high temporal resolution. Its design principles could also serve as a blueprint for the development of similar, orthogonal biosensors for other signaling molecules, metabolites and antibiotics.

The ubiquitous second messenger c-di-GMP plays pivotal roles in many bacteria, regulating behavioral and physiological processes like motility and virulence, or adherence to and growth on surfaces[1–8]. Accurate control of c-di-GMP was shown to be critical for the establishment of infections and the development of resilience against host-mediated stress and antibiotic therapy in several human pathogens[9]. In most bacteria, regulatory networks controlling c-di-GMP are complex with a multitude of enzymes being responsible for the controlled synthesis and degradation of the signaling compound[10–12]. For example, the human pathogen *Pseudomonas aeruginosa* encodes 38 enzymes involved in the "make and break" of c-di-GMP[13], and in some phyla enzymes regulating c-di-GMP constitute >1% of the organism's total protein repertoire[1]. This makes the genetic dissection of the respective regulatory pathways and mechanisms challenging. Much of the information on c-di-GMP control and its role in bacterial physiology and behavior stems from extrapolations of genetic data, generally complemented with invasive biochemical assays providing snapshots of c-di-GMP at steady-state levels and in bulk populations. Because such measurements do not provide single cell information, they fail to visualize signaling heterogeneity in bacterial populations, track dynamic fluctuations in asynchronous populations, or distinguish distinct c-di-GMP-mediated cell fates in bacterial communities. Advancing the mechanistic understanding of c-di-GMP signaling in bacteria thus requires genetically encoded biosensors that monitor dynamic changes of c-di-GMP levels in real-time and in individual cells in a non-invasive manner.

The strong interest in robust biosensors for c-di-GMP or other small molecules has inspired repeated attempts to develop such tools. However, available biosensors generally suffer from major drawbacks. Transcription- or translation-based reporters, although readily available, are generally not suitable for real-time measurements due to considerable temporal delays caused by expression, folding,

[1]Biozentrum, University of Basel, Spitalstrasse 41, 4056 Basel, Switzerland. ✉e-mail: andreas.kaczmarczyk@unibas.ch; urs.jenal@unibas.ch

maturation or stability of the reporters[14–19]. This limitation can be overcome by allosteric sensors operating at the posttranslational level. For instance, a recently described sensor based on bimolecular fluorescence complementation (BiFC) is suitable to measure steady-state levels of c-di-GMP[20]. However, the irreversible nature of reconstituting a functional fluorescent protein by BiFC[21] precludes this tool from accurately measuring dynamic c-di-GMP fluctuations. More suitable biosensors that can potentially track changes in c-di-GMP levels in real time make use of Förster resonance energy transfer (FRET) between two fluorescent proteins[22–24], or bioluminescence resonance energy transfer (BRET) where a ligand-binding protein is linked to a luciferase and a fluorescent protein[25]. While FRET sensors can identify and monitor individual cells with different levels of c-di-GMP, they generally lack robustness and, so far, have not been shown to record dynamic changes of c-di-GMP over physiologically relevant time scales. In addition, FRET sensors generally suffer from a limited dynamic range, making them highly susceptible to noise and imposing challenging downstream analysis procedures.

Here we set out to develop a biosensor that overcomes the above limitations and allows monitoring of c-di-GMP dynamics in a diverse range of bacteria by accessing several single-cell techniques, including live-cell microscopy and flow cytometry. Starting from a circularly permuted enhanced green fluorescent protein (cpEGFP) scaffold sandwiched by c-di-GMP-binding domains, we apply a directed evolution-inspired approach[26,27], where c-di-GMP-responsive biosensors with increasing dynamic range and rapid binding kinetics are gradually selected by using iterative fluorescence-activated cell sorting (FACS) under alternating c-di-GMP regimes. The resulting single fluorescent protein biosensor (SFPB), termed cdGreen2, is benchmarked by dissecting the c-di-GMP regulatory networks of two model organisms, the environmental bacterium *Caulobacter crescentus* and the human pathogen *Pseudomonas aeruginosa*. The bimodal developmental program of *C. crescentus* generates a motile and planktonic swarmer (SW) cell and a sessile, surface-attached stalked (ST) cell. Cell differentiation was proposed to depend on precise and genetically hardwired oscillations of c-di-GMP that coordinate cell cycle progression with morphogenesis and behavior[3,19]. Similarly, *P. aeruginosa* undergoes a surface-induced asymmetric program generating motile and sessile offspring to maximize surface colonization[4,28]. Similar to *C. crescentus*, this program was postulated to be orchestrated by the asymmetric distribution of c-di-GMP during the initial cell divisions of surface-adherent *P. aeruginosa* cells. Using cdGreen2 to visualize c-di-GMP in individual cells, we corroborate these models and define the molecular mechanisms underlying the bimodal phenotypic specialization. We anticipate that cdGreen2 will become a standard molecular tool for the scientific community to dissect the function and mechanisms of c-di-GMP in a large variety of bacteria.

## Results

### Design and directed evolution of a c-di-GMP-specific biosensor

To develop a biosensor for c-di-GMP, we generated libraries with circularly permutated EGFP (cpEGFP)[29] sandwiched by two c-di-GMP-binding C-terminal domains (CTDs) of the transcription factor BldD (BldD$_{CTD}$)[30] (Fig. 1a). Circular permutation places the new termini of cpEGFP close to the chromophore such that its optical properties are highly sensitive to the immediate microenvironment[31–35]. We chose BldD because it is monomeric in the absence of c-di-GMP, but dimerizes in the presence of the ligand with two intercalated dimers of c-di-GMP bridging the BldD protomers[30] (Fig. 1b, c). Libraries were generated by introducing variations in length and rigidity/flexibility of the linkers connecting BldD moieties and cpEGFP (Supplementary Fig. 1a, b). Linker rigidity/flexibility was varied by introducing Pro or Ser [PS] residues and Ala or Gly [AG] residues at alternating positions, since Gly-Ser-rich linkers are flexible, whereas Pro-Ala repeats provide stiffness to linkers[36]. To further diversify the library, degenerate

codons were introduced at select positions to add charged residues like Lys, Arg or Glu to the linker regions (Supplementary Fig. 1a, b). The linker library was then transferred into an *Escherichia coli* strain (AKS494) (Fig. 1d), in which the concentration of c-di-GMP could be adjusted from <50 nM to >5 μM[37]. Next, iterative fluorescence-activated cell sorting (FACS) under alternating regimes of high and low c-di-GMP concentration was used to enrich for functional biosensors displaying maximal differences in fluorescence intensity between the two c-di-GMP states (Fig. 1d). This led to the isolation of a first-generation biosensor with approximately three-fold change in fluorescence intensity upon expression in *E. coli* cells with low and high c-di-GMP concentrations, respectively (Fig. 1e). Sequential cycles of linker optimization and selection (see Methods) led to gradually improved versions of the biosensor, generating a final variant, termed cdGreen, with a more than ten-fold increase in fluorescence intensity in the presence of c-di-GMP (Fig. 1e; Supplementary Fig. 1c, d).

Because SFPBs are intensiometric by design and their readout prone to naturally occurring variations of protein concentration in vivo, we sought to normalize c-di-GMP-mediated changes in fluorescence intensity by fusing the sensor to a reference fluorescent protein (FP). To avoid interference of the reference FP with biosensor functionality, we followed the recently developed "Matryoshka" approach[38], inserting mScarlet-I[39] in the loop connecting the original EGFP N- and C-termini (Fig. 1f, g). The resulting construct, termed cdGreen-Matry, retained its c-di-GMP sensitivity in vivo (Fig. 1h), making it possible to normalize the cpEGFP-based c-di-GMP signal with the reference FP (Fig. 1i), thus effectively converting the initial intensiometric into a ratiometric biosensor.

### cdGreen is a ratiometric c-di-GMP biosensor with high ligand specificity and dynamic range

To demonstrate that cdGreen specifically responds to c-di-GMP, we interchanged the Arg and Asp residues of ligand-binding motif 2 in one of the BldD$_{CTD}$ protomers (Fig. 1c, g). These substitutions were previously shown to fully abrogate c-di-GMP binding in the context of full-length BldD[30,40]. Accordingly, the mutant sensor failed to respond to c-di-GMP (Fig. 1h, i). Next, we expressed and purified hexa-His-tagged cdGreen and cdGreen-Matry to homogeneity (Supplementary Fig. 2a). Purified cdGreen showed two major excitation peaks at 497 nm and 405 nm with corresponding emission maxima at 513 and 518 nm, respectively (Fig. 2a). Importantly, addition of c-di-GMP strongly increased the excitation peak at 497 nm, but reduced the peak at 405 nm in a dose-dependent manner with concomitant changes of the corresponding emission peaks (Fig. 2b). The inverse response of two major excitation peaks to c-di-GMP provides a ratiometric readout, a feature critical for in vivo applications as it provides an elegant way to normalize stochastic cell-to-cell variations of biosensor concentrations. Purified cdGreen-Matry showed a similar c-di-GMP-dependent behavior, but a reduced response amplitude due to Förster resonance energy transfer (FRET) between its cpEGFP and mScarlet-I moieties (Fig. 2a). The dose-response curves recorded for cdGreen upon excitation at 497 nm and 405 nm and emission at 530 nm revealed sigmoidal monotonic behavior for both wavelengths with maximal fold changes of 12.0 and 5.0 and fitted $K_d$ values of 214 nM (95% CI: 210–217 nM; Hill slope = 2.30) and 225 nM (95% CI: 217–228 nM; Hill slope = −2.43), respectively (Fig. 2c). The combined ratiometric readout also displayed a sigmoidal and monotonic behavior with a dynamic range of almost 70 and a fitted $K_d$ of 386 nM (95% CI: 381–391 nM; Hill slope: 3.31) (Fig. 2c). The response of purified cdGreen was highly specific to c-di-GMP, as none of the related nucleotides tested induced a response or caused interference with the binding of c-di-GMP (Supplementary Fig. 3a). Stoichiometric titration experiments revealed a plateau at a ratio of two c-di-GMP molecules per one molecule of protein (Supplementary Fig. 3b). This is in line with the observation that BldD, although binding two intercalated c-di-GMP dimers in the

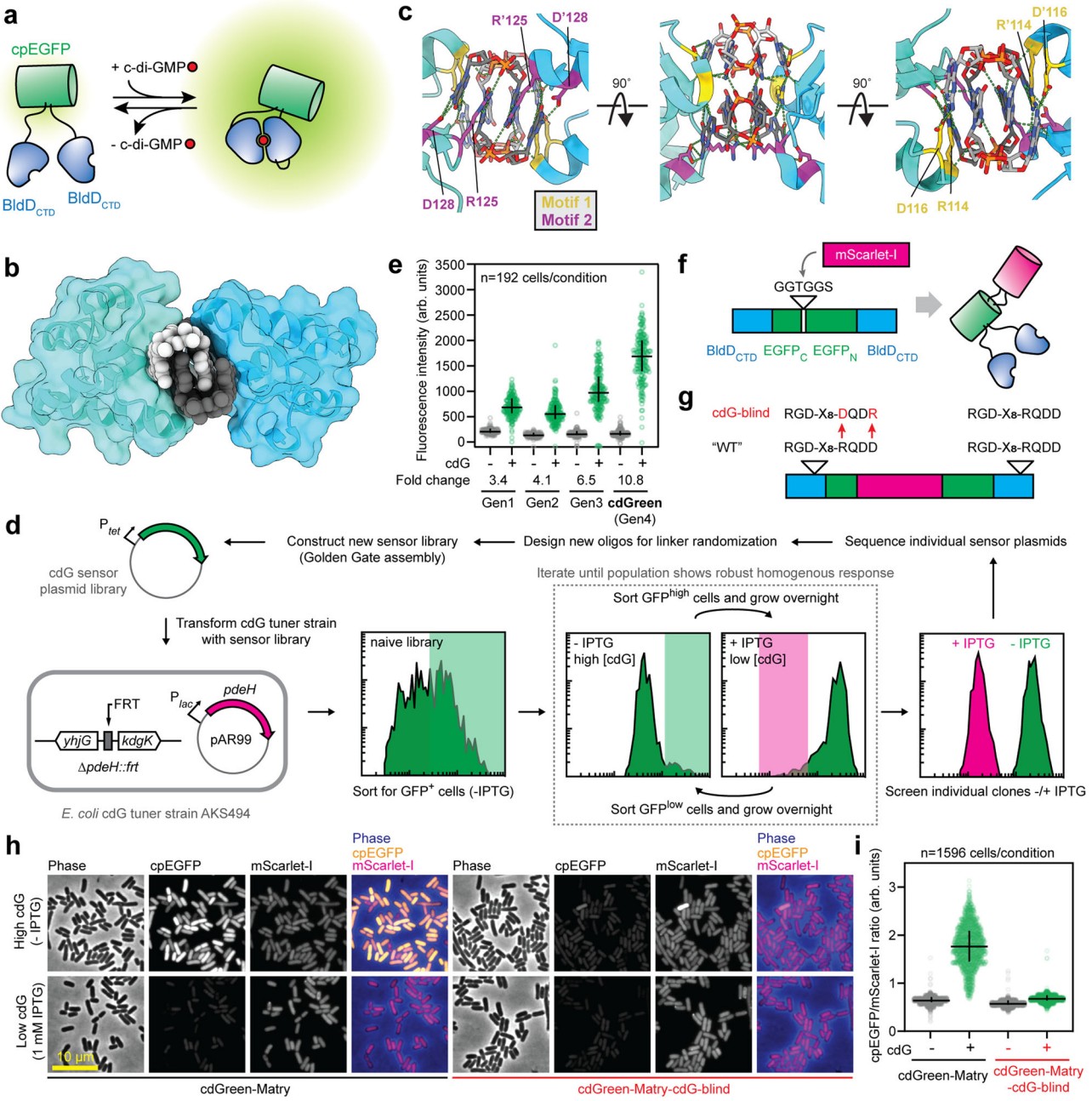

**Fig. 1 | Design and isolation of c-di-GMP biosensor cdGreen. a** Schematic of cdGreen biosensor design and function. Upon c-di-GMP-mediated dimerization of the two BldD$_{CTD}$ protomers, the cpEGFP moiety undergoes a conformational change resulting in increased fluorescence intensity. **b** Cartoon representation of the dimeric BldD$_{CTD}$ in complex with tetrameric c-di-GMP (PDB: 5KHD). The intercalated c-di-GMP dimer binding to motif 1 is shown with individual c-di-GMP molecules in light and dark grey, respectively. Note that dimerization is solely mediated by c-di-GMP without any contribution from protein-protein interactions. **c** Close-up of the BldD$_{CTD}$-c-di-GMP interface with key residues (underlined) in motif 1 (RGD) and motif 2 (RQDD) highlighted as yellow and purple sticks, respectively. Based on PDB: 5KHD. **d** Schematic outline of the iterative FACS approach used to isolate c-di-GMP biosensors, including the 4th-generation biosensor cdGreen (see Methods for details). **e** Performance of different c-di-GMP

biosensors in c-di-GMP high and low conditions assayed in strain AKS494 by microscopy. For "c-di-GMP low" conditions, cells were grown with 1 mM IPTG. 200 nM aTc was included in all conditions for expression of the biosensors. For image analysis, images were background-corrected. Medians and interquartile ranges are indicated. **f** Schematic of the cdGreen-Matry construct. **g** Schematic of the cdG-blind cdGreen-Matry construct. **h** Microscopy snapshots of AKS494 harboring plasmids p2H12-Matry or p2H12-Matry-cdG-blind grown with (1 mM) or without IPTG. 200 nM aTc was included in all conditions for expression of the biosensors. Representative images from two independent biological replicates are shown. **i** Quantification of c-di-GMP levels (cpEGFP/mScarlet-I ratio) of strains shown in panel **f**. Medians and interquartile ranges are indicated. arb. units, arbitrary units; cdG, c-di-GMP. Source data are provided as a Source Data file.

fully functional state (Fig. 1b, c), is able to dimerize with only one c-di-GMP dimer bound[40]. Isothermal titration calorimetry (ITC) experiments showed two consecutive binding events of c-di-GMP to cdGreen, likely corresponding to dimeric and tetrameric c-di-GMP (Supplementary Fig. 3c). These results demonstrated that cdGreen

binds tetrameric c-di-GMP and that one intercalated c-di-GMP dimer is sufficient to induce the maximal response.

Next, we sought to demonstrate that the ratiometric readout of cdGreen can also be used in vivo. Indeed, fluorescence measurements of *E. coli* strain AKS494 expressing cdGreen (Fig. 1d) by flow cytometry

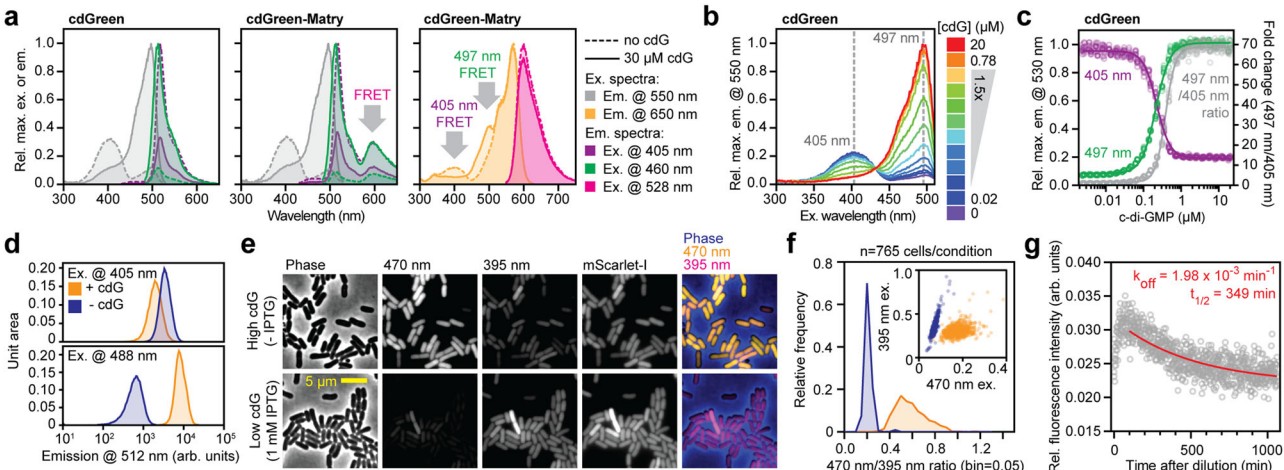

**Fig. 2 | In vitro characterization of c-di-GMP biosensors cdGreen and cdGreen-Matry. a** Spectral excitation and emission scans of cdGreen and cdGreen-Matry. Peaks resulting from FRET between cpEGFP and mScarlet-I in cdGreen-Matry are indicated. **b** Excitation spectra of cdGreen with different c-di-GMP concentrations. **c** C-di-GMP dose-response curve of cdGreen emission at 530 nm with excitation at 405 nm and 497 nm (left y-axis) and the combined ratiometric readout (ratio of 550 nm emission signal with 497 nm and 405 nm; right *y*-axis). Pooled data from two replicates with repeated measurements (see Methods). **d** Flow cytometry of

strain AKS494 harboring plasmid p2H12ref grown with (1 mM) or without IPTG. 200 nM aTc was included in all conditions for expression of the biosensors. **e** Microscopy images of strain AKS494 carrying plasmid pConRef-2H12 grown with (1 mM) or without IPTG. **f** Quantification of c-di-GMP levels (470 nm/395 nm ratio) of strains shown in panel **e**. **g** Dissociation kinetics of cdGreen as determined by the "dissociation by dilution" method (for details, see Methods). Pooled data from triplicates ($n = 3$). arb. units, arbitrary units; cdG, c-di-GMP. Source data are provided as a Source Data file.

(Fig. 2d) or live-cell imaging (Fig. 2e, f) at high or low c-di-GMP concentrations readily detected signals in the GFP/FITC emission channel upon excitation at 395–405 nm, with signal strengths anti-correlating with the GFP/FITC emission channel signals upon excitation with the standard GFP/FITC excitation wavelengths (470–488 nm). Excitation around 400 nm is routinely used for DAPI excitation, making standard settings in most modern fluorescence microscopes and flow cytometry instruments ideally suited to adopt the c-di-GMP biosensor cdGreen without the need of specialized equipment.

## Isolation of cdGreen2, a dynamic and robust ratiometric c-di-GMP biosensor with rapid off kinetics

To evaluate the potential of cdGreen to read out dynamic changes of c-di-GMP in vivo, we next determined the dissociation rate constant ($k_{off}$) using the "dissociation by dilution" method[41] (for details, see Methods). This revealed a $k_{off}$ of $1.98 \times 10^{-3}$ min$^{-1}$ (95% CI: 1.62–2.36 $\times 10^{-3}$ in$^{-1}$) corresponding to a half-life of 349 min (95% CI: 294–428 min) (Fig. 2g). Based on the experimentally determined $k_{off}$ and $K_d$ values, we calculated an association rate constant ($k_{on} = k_{off} K_d^{-1}$) of around 9000 M$^{-1}$ min$^{-1}$. Thus, although the dynamic range of cdGreen is exceptionally high, slow dissociation of c-di-GMP likely precludes dynamic measurements in vivo. To test this, we expressed cdGreen in *Caulobacter crescentus*, an aquatic bacterium with a characteristic asymmetric division cycle generating motile and sessile offspring. While levels of c-di-GMP are high in sessile stalked (ST) cells, newborn swarmer (SW) cells experience an extended trough of c-di-GMP during their motile phase, before the concentration of c-di-GMP sharply rises to drive morphogenesis into sessile replication-competent ST cells (Fig. 3a)[42,43]. In line with the slow off kinetics measured in vitro, cdGreen expressed in *C. crescentus* reported constant high levels of c-di-GMP but failed to reproduce the expected fluctuations of the ligand during the cell cycle (Supplementary Fig. 4a). When cdGreen was expressed in a *C. crescentus* mutant unable to produce c-di-GMP (NA1000 rcdG$^0$)[42], a signal corresponding to low c-di-GMP levels was observed (Supplementary Fig. 4b). This suggested that while cdGreen is fully functional in *C. crescentus*, it is unable to read out dynamic fluctuations of c-di-GMP, likely due to its slow dissociation kinetics.

Because the affinity equilibrium dissociation constant $K_d$ directly relates to the dissociation and association rate constants $k_{off}$ and $k_{on}$ ($K_d = k_{off} k_{on}^{-1}$), we reasoned that variants with lower binding affinities may show more rapid off kinetics. We thus generated mutant libraries with randomized amino acids at non-conserved positions in the immediate vicinity of residues directly involved in c-di-GMP binding (Figs. 1c, 3b) and subjected them to iterative FACS-based enrichment for variants that shift to the off state at higher c-di-GMP concentrations as compared to cdGreen (see Methods). Two variants were isolated by this selection, 2H12-Aff2 and 2H12-Aff8, and both had Pro substitutions in the original RQDD ligand binding motif 2 (RPAD and RPPD, respectively) in addition to non-common mutations in motif 1 (Figs. 1c, g and 3b). We speculate that the Pro substitutions reposition residues R125 and D128 in a way that c-di-GMP is bound less tightly compared to the wild-type binding site in cdGreen. Because 2H12-Aff2 and 2H12-Aff8 still failed to recapitulate expected c-di-GMP oscillations in *C. crescentus* (Supplementary Fig. 5a), additional cdGreen-based derivatives were constructed by combining motif 1 and/or motif 2 mutations of 2H12-Aff2 and 2H12-Aff8 variants in one or both BldD$_{CTD}$ protomers. Scoring the resulting variants (see Methods) led to the isolation of a single biosensor, termed cdGreen2, that harbored the RPAD mutant motif in both BldD$_{CTD}$ protomers and recapitulated the expected c-di-GMP oscillations over the *Caulobacter* cell cycle (Supplementary Fig. 5a, b). Cells expressing cdGreen2 displayed constantly high levels of c-di-GMP in their ST phase, while levels dropped in newborn SW cells for a period of 20–25 min, before reaching their original levels upon SW-to-ST cell differentiation (Fig. 3c). Co-expression of cdGreen2 with a ST cell-specific marker (SpmX-mCherry)[19,44] confirmed that the drop in c-di-GMP coincided with the motile SW stage of the *Caulobacter* cell cycle (Supplementary Fig. 5c). Importantly, cdGreen2 robustly imaged c-di-GMP fluctuations over several hours and generations without decrease in sensor performance (Supplementary Fig. 5d).

Purified cdGreen2 showed spectral properties similar to parental cdGreen with major excitation peaks at 497 nm and 405 nm and with corresponding emission maxima at 513 and 518 nm, respectively (Fig. 3d). Addition of c-di-GMP led to increased fluorescence upon excitation at 497 nm and decreased emission upon excitation at

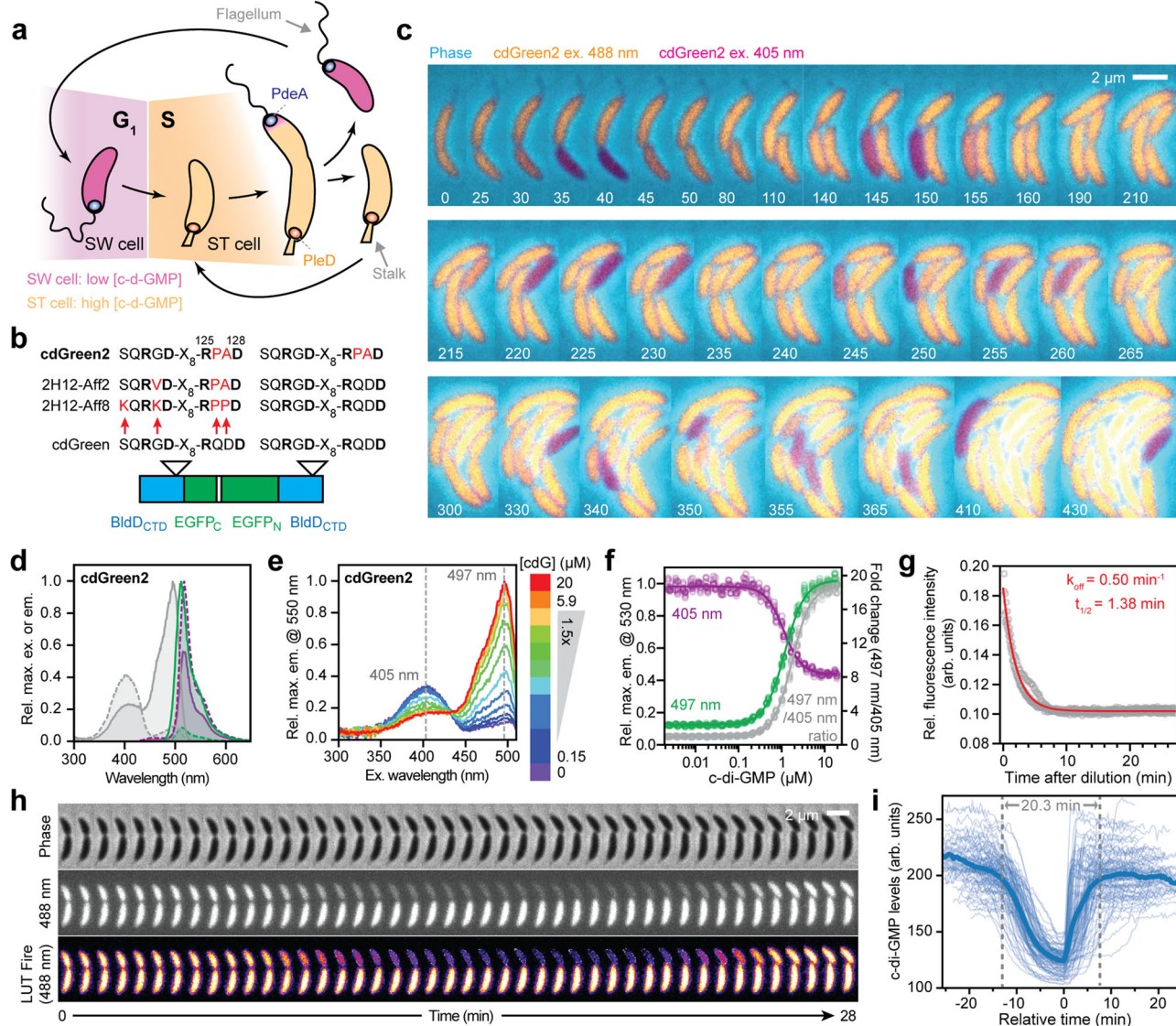

**Fig. 3 | Isolation and in vitro characterization of c-di-GMP biosensor cdGreen2.**
**a** Schematic of c-di-GMP oscillations during the *C. crescentus* cell cycle. Stalk and flagellum are indicated, as well as the opposite polar localization of the major diguanylate cyclase PleD at the stalked pole and the major phosphodiesterase PdeA at the incipient flagellated pole. **b** Schematic of amino acid changes in cdGreen derivatives 2H12-Aff2, 2H12-Aff8 and cdGreen2. **c** Time-lapse series of strain NA1000 carrying plasmid pQFmcs-2H12.D11. Time stamps indicate minutes. Images are overlays of the cdGreen2 FITC/FITC signal pseudo-colored in yellow, the cdGreen2 DAPI/FITC signal pseudo-colored in magenta and the phase contrast image pseudo-colored in cyan. Also see related Supplementary Fig. 5d. **d** Spectral excitation and emission scans of cdGreen2. Emission and excitation wavelengths are color-coded as in Fig. 2a. **e** Excitation spectra of cdGreen2 with different c-di-GMP concentrations. **f** C-di-GMP dose-response curve of cdGreen2 emission at 530 nm with excitation at 405 nm and 497 nm (left *y*-axis) and the combined ratiometric readout (ratio of 550 nm emission signal with 497 nm and 405 nm; right *y*-axis). Pooled data from two replicates with repeated measurements (see Methods). **g** Dissociation kinetics of cdGreen2 as determined by the "dissociation by dilution" method (for details, see Methods). Pooled data from triplicates ($n = 3$). **h** High-temporal-resolution imaging of c-di-GMP levels (20-s intervals) during *C. crescentus* division and G1-S phase transition. Note that only every second frame is shown. **i**, Quantification of single-cell c-di-GMP dynamics with high temporal resolution ($n = 96$ cells). Individual cell tracks were overlaid, setting the time point at which a cell showed minimum c-di-GMP levels as a reference point ($t = 0$). arb. units, arbitrary units; cdG, c-di-GMP. Source data are provided as a Source Data file.

405 nm in a dose-dependent manner (Fig. 3d, e). Dose-response curves at 497 nm and 405 nm excitation showed sigmoidal monotonic behavior with maximal fold changes of 8.2 and 2.2 and with fitted $K_d$ values of 1.24 μM (95% CI: 1.22–1.26 μM; Hill slope = 1.86) and 1.17 μM (95% CI: 1.12–1.21 μM; Hill slope = −2.07), respectively. The combined ratiometric readout displayed a sigmoidal and monotonic behavior with a maximal fold change of almost 20 and a fitted $K_d$ of 1.83 μM (95% CI: 1.81–1.85 μM; Hill slope: 1.98) (Fig. 3f). Stoichiometric titration experiments revealed a 2:1 ligand-to-protein ratio, similar to parental cdGreen (Supplementary Fig. 3d). Finally, "dissociation by dilution" experiments[41] with purified cdGreen2 determined a dissociation rate constant ($k_{off}$) of 0.50 min⁻¹ (95% CI: 0.49–0.52 min⁻¹) corresponding to

a half-life of 1.38 min (95% CI: 1.33–1.43 min) (Fig. 3g). The calculated association rate constant ($k_{on} = k_{off} K_d^{-1}$) is around 400,000 M⁻¹ min⁻¹. These kinetic values were confirmed by independent experiments globally fitting parameters based on association kinetics in the pseudo-first-order regime (see Methods) (Supplementary Fig. 3e, f).

Time-lapse microscopy of dividing *C. crescentus* cells expressing cdGreen2 were recorded at 20-s intervals, exposing a robust c-di-GMP trough period of about 20 min in differentiating SW cells (Fig. 3h, i). Importantly, no phototoxicity or bleaching effects were observed when cells were imaged with high temporal resolution for up to 24 h (also see Supplementary Movie 1). This exceptional robustness likely relates to the overall signal strength of cdGreen2, which allows the use

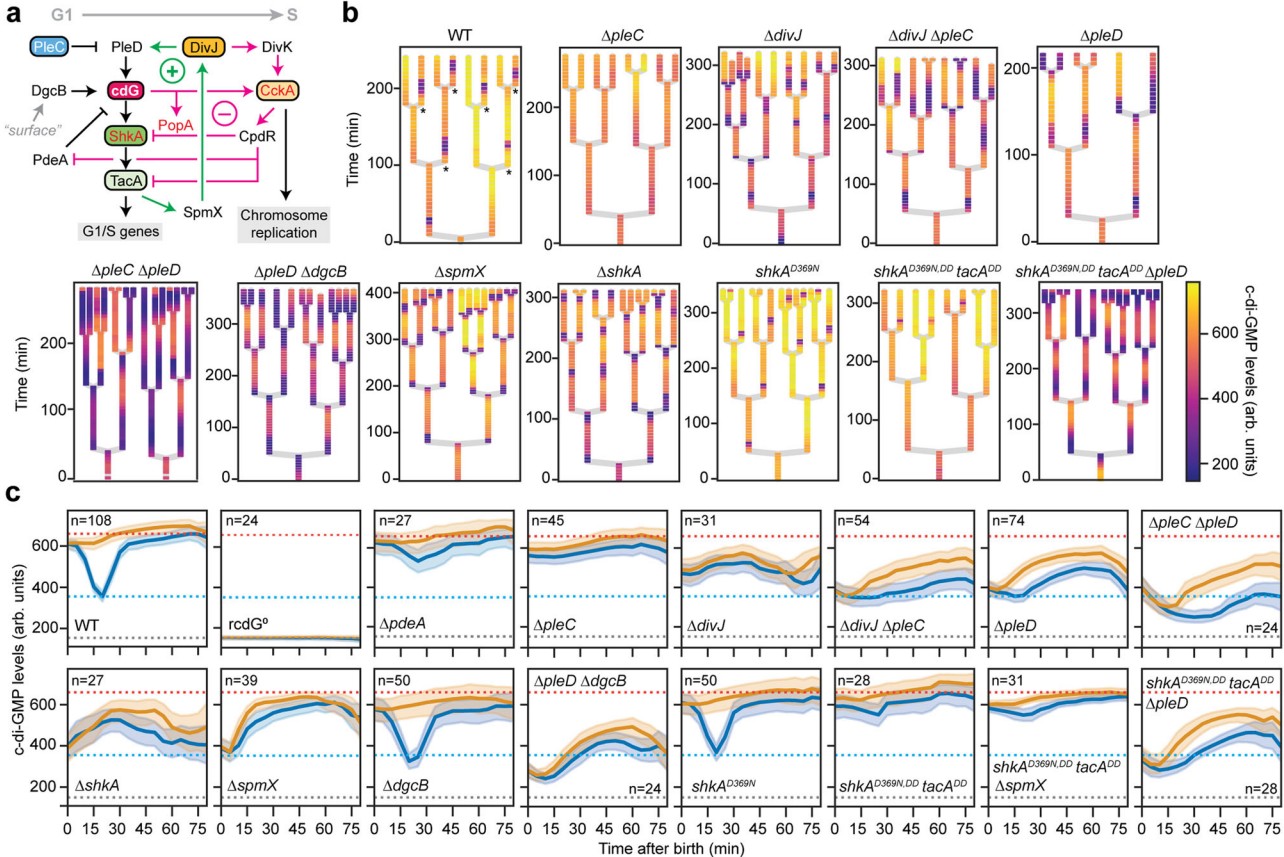

**Fig. 4 | Dissecting the G1-S-specific c-di-GMP network of *C. crescentus*.**
**a** Schematic of the regulatory network driving G1-S transition in *C. crescentus*. C-di-GMP effector proteins, i.e., proteins whose functions are directly controlled by c-di-GMP binding, are highlighted in red. CpdR and PopA are ClpX-specific adapter proteins required for ShkA, TacA and/or PdeA proteolysis; PopA activity requires c-di-GMP binding. The positive and negative feedback loops (see main text for details) during G1-S phase transition are highlighted by green and magenta arrows, respectively. **b** Lineage trees of indicated strains carrying plasmid pQFmcs-2H12.D11-scarREF. Asterisks in the "wild type" panel indicate SW cells; note that ST cells show shorter interdivision times than SW cells. **c** Tracking of c-di-GMP

levels in pairs of sister cells after cell division (*n* = number of pairs comprising one c-di-GMP-high and one c-di-GMP-low cell; values for each strain are indicated in the figure). Means (solid lines) and 95% confidence intervals (shaded error bands) for c-di-GMP-high (orange) and c-di-GMP-low (blue) sister cells are shown. The orange dotted line indicates average c-di-GMP levels in wild-type ST cells, the blue dotted line indicates average minimal c-di-GMP levels in wild-type SW cells; the grey dotted line indicates average c-di-GMP levels in rcdG⁰ cells and thus reflects the background signal in absence of c-di-GMP. arb. units, arbitrary units; cdG, c-di-GMP. Source data are provided as a Source Data file.

of short exposure times in the fluorescent channels and reduced laser power during time-lapse experiments. In sum, these results demonstrate that cdGreen2 affords robust, quantitative and dynamic monitoring of c-di-GMP at the single-cell level with high temporal resolution and without phototoxic effects or photobleaching over multiple generations.

## cdGreen2 unveils the regulatory network driving *Caulobacter* morphogenesis

Accurate temporal and spatial control of c-di-GMP was proposed to be a central element of *C. crescentus* morphogenesis and cell cycle progression[19]. Synthesis and degradation of c-di-GMP are controlled by two key enzymes, the diguanylate cyclase PleD and the phosphodiesterase PdeA, which position to opposite poles of dividing cells and differentially partition into ST and SW progeny during division[3] (Figs. 3a, 4a). To investigate the contribution of these enzymes to spatial and temporal control of c-di-GMP in individual cells, we used cdGreen2 to visualize c-di-GMP over multiple generations and to generate lineage trees originating from single dividing cells (Fig. 4b). Analyzing multiple sister cell pairs after division underscored the highly deterministic nature of c-di-GMP asymmetry during the bimodal *Caulobacter* cell cycle (Fig. 4c). Genetic evidence had suggested that c-di-GMP levels are kept low in SW cells by PdeA[45] and by the

phosphatase PleC, which retains PleD in its inactive, unphosphorylated state[46–49] (Fig. 4a). In line with this, c-di-GMP failed to drop in newborn SW cells of mutants lacking PdeA or PleC (Fig. 4b, c; Supplementary Fig. 6a, b). Deleting *pleD* in the Δ*pleC* mutant lowered levels of c-di-GMP and essentially phenocopied a Δ*pleD* single mutant (see below), demonstrating that PleD is indeed overactivated in the absence of the SW cell-specific phosphatase PleC (Fig. 4b, c). Levels of c-di-GMP were significantly reduced in a Δ*pleD* mutant, and were even lower in a strain that also lacked DgcB, a second important diguanylate cyclase which promotes rapid surface attachment of SW cells in response to mechanical cues[50]. Although a Δ*pleD* Δ*dgcB* double mutant showed strongly reduced levels of c-di-GMP levels, the concentration of c-di-GMP still changed over the cell cycle, indicating that additional, minor diguanylate cyclases are involved in this process[42]. However, c-di-GMP levels in SW and ST cell progeny were indistinguishable in the double mutant (Fig. 4b), indicating that PleD and DgcB are indeed the principal drivers of c-di-GMP asymmetry in *Caulobacter*.

As *Caulobacter* cells transit from G1 into S phase, c-di-GMP levels gradually increase[42,43] after PleD is activated by phosphorylation[49] and PdeA is proteolytically removed[45] (Fig. 4a). This leads to a series of accurately timed events prompting exit from G1, SW-to-ST cell morphogenesis and entry into S phase. First, c-di-GMP-dependent activation of the ShkA-TacA pathway stimulates the expression of hundreds

of genes that orchestrate the morphological restructuring of the motile SW cell into the sessile ST cell[19,51–53]. This sets in motion a positive feedback loop that includes the kinase DivJ and its activator SpmX, leading to the production of more c-di-GMP via the reinforced activation of PleD (Fig. 4a). The resulting buildup of c-di-GMP was proposed to trigger a second, negative feedback loop via the cell cycle kinase CckA that terminates ShkA-mediated gene expression and initiates chromosome replication upon entry into S-phase[19,51,54–57]. While this complex regulatory network was proposed to accurately time *Caulobacter* development (Fig. 4a), direct visualization of c-di-GMP in support of the model is missing. We now show that mutants lacking components of the proposed positive feedback loop (ShkA, SpmX, DivJ) all showed reduced c-di-GMP levels. For example, a Δ*divJ* mutant failed to increase the concentration of c-di-GMP during G1-S transition and, similar to a Δ*pleD* mutant, showed strongly compromised c-di-GMP asymmetry during cell division (Fig. 4b, c). A Δ*divJ* Δ*pleC* double mutant largely phenocopied a Δ*divJ* mutant with respect to c-di-GMP dynamics, suggesting that PleD remains mostly unphosphorylated and inactive in this background. Mutants lacking ShkA or SpmX showed similar phenotypes, supporting their proposed role in robustly boosting c-di-GMP levels during the G1-S transition (Fig. 4b, c). Interestingly, Δ*pleD*, Δ*divJ*, Δ*spmX* and Δ*shkA* single mutants, as well as the Δ*divJ* Δ*pleC* double mutant, showed a premature trough of c-di-GMP in late predivisional cells, which extended into both daughter cells after division. This observation indicated that (i) PdeA and possibly other phosphodiesterases are active already before cell constriction, and that (ii) robust activation of PleD is important to overrule the phosphodiesterase(s) and to maintain cellular identity of ST cells.

Interference with components of the negative feedback loop had the opposite effect. Introducing amino acid substitutions in ShkA and TacA that protected them from being degraded (*shkA^{DD}* *tacA^{DD}*)[19,58] in combination with a mutation that rendered ShkA constitutively active (*shkA^{D369N}*)[19] led to constitutively high levels of c-di-GMP, bypassing the characteristic drop of c-di-GMP in SW cells (Fig. 4b, c). Deletion of *pleD* in the *shkA^{D369N,DD}* *tacA^{DD}* background essentially phenocopied a Δ*pleD* mutant, confirming that the negative feedback loop mediating ShkA and TacA proteolysis limits PleD-mediated activation of the ShkA-TacA pathway to a short time window, thereby providing accurate gene expression control during the G1-S transition. Deletion of *spmX* in the *shkA^{D369N,DD}* *tacA^{DD}* background had no such effect, suggesting that other direct or indirect ShkA/TacA-dependent factors stimulate PleD activity. Together, these results provide strong support for a developmental model in which the gradual upshift of c-di-GMP and the precise timing of gene expression during the *C. crescentus* cell cycle is executed by consecutive positive and negative feedback loops. Our findings also demonstrate that accurate control of the diguanylate cyclase PleD is of key importance to establish c-di-GMP asymmetry in *C. crescentus*. Finally, these experiments demonstrate the potential of the cdGreen2 biosensor as a powerful cellular tool to directly correlate cellular dynamics and specific gene functions with accurate measurements of a small signaling molecule in individual cells.

## Recording c-di-GMP during *Pseudomonas aeruginosa* surface colonization

To demonstrate the versatility of the biosensor, we adopted cdGreen2 to study *Pseudomonas aeruginosa*, an opportunistic human pathogen that is notorious for its capacity to colonize and breach mucosal linings and cause acute and chronic infections. We have recently uncovered a sophisticated asymmetric program termed "touch-seed-and-go" that allows *P. aeruginosa* to effectively colonize surfaces[28]. We had proposed that upon surface contact *P. aeruginosa* senses mechanical cues with its polar rotary flagellar motor and, in response, triggers a rapid upshift of c-di-GMP to stimulate surface attachment via the production of adhesive Type IV pili[59]. Surface attached cells then undergo an

asymmetric division, generating a surface-adherent offspring that retains high levels of c-di-GMP, and an offspring that reduces its internal c-di-GMP to regain motility and explore distant sites[28]. The reduction of c-di-GMP in one offspring depends on the activity of the phosphodiesterase Pch, which positions asymmetrically during division by associating with the chemotaxis machinery at the flagellated pole[60] (Fig. 5a). Using a codon-adapted version of cdGreen2, called cdGreen2.1 (Supplementary Data 1), we set out to scrutinize the proposed *P. aeruginosa* surface colonization model. Because under certain growth conditions, *P. aeruginosa* autofluorescence complicates internal normalization using the 405 nm excitation peak of cdGreen2, we included a transcriptionally coupled reference fluorophore (mScarlet-I) for normalization (Fig. 5b).

Using cdGreen2.1, c-di-GMP was readily detected in *P. aeruginosa* strain PAO1 grown on LB agarose pads. As predicted by the "touch-seed-and-go" model, the first few division events on surface generated offspring with asymmetric c-di-GMP distributions (Fig. 5c). This was followed by a gradual increase of c-di-GMP heterogeneity with an increasing fraction of newborn cells adopting a low c-di-GMP state, arguing that genetic and environmental factors gradually influence the c-di-GMP program upon surface colonization (Fig. 5c). A strain lacking the phosphodiesterase Pch had lost its asymmetry, displaying uniformly high c-di-GMP for an extended period of time (Fig. 5c). In line with the proposed role of the polar flagellum in sensing mechanical cues upon surface contact[28,50], a strain lacking the motor components MotA and MotB failed to increase c-di-GMP levels throughout the first few divisions on surface. Cell-to-cell variability of c-di-GMP was still observed in this strain, although at strongly reduced levels, indicating that pathways independent of flagellar-based mechanosensation contribute to an upshift of c-di-GMP under these conditions (Fig. 5c). Finally, we tested the role of RetS in c-di-GMP control, an atypical histidine kinase in the Gac/Rsm pathway[61–63] that was reported to negatively influence c-di-GMP levels in *P. aeruginosa* strain PAK[7]. A Δ*retS* mutant of strain PAO1 showed strongly increased levels of c-di-GMP even at very early time points of surface exposure. This and the observation that the characteristic asymmetric distribution of c-di-GMP was delayed in this mutant (Fig. 5b, d), argued that the role of the Gac/Rsm pathway is to limit c-di-GMP in planktonic *P. aeruginosa* cells, providing them with the necessary sensitivity to respond to surface encounters by rapidly boosting c-di-GMP levels and leading to effective surface adherence via c-di-GMP-mediated Type IV pili assembly[28]. While the surface-mediated boost and subsequent asymmetry of c-di-GMP was also highly pronounced in *P. fluorescens* Pf0-1, an environmental *Pseudomonas* isolate, *P. aeruginosa* strains PA14 and PAK showed only weak responses to surface exposure (Fig. 5d, e), suggesting that they adapt to surfaces differently than strain PAO1. Given the notion that c-di-GMP often counters virulence of *P. aeruginosa* and other bacterial pathogens[1], it is possible that differential c-di-GMP control explains the hyper-virulent phenotype of strain PA14[64,65]. In this context, it is worth noting that the hypervirulence of strain PA14 has previously been linked to a mutation in *ladS*, encoding a kinase affecting Gac/Rsm pathway activity[66].

Altogether, these examples demonstrate the versatility and adaptability of the cdGreen2 biosensor for the detailed analysis of a diverse range of bacteria. Its ability to robustly and sensitively report on c-di-GMP dynamics in individual bacteria and within the physiological concentration of the signaling molecule underscores its potential value for the dissection of signal transduction networks in vivo.

## Use of cdGreen2 to monitor c-di-GMP turnover in vitro

Because c-di-GMP is the regulatory mastermind of biofilm formation in most bacteria and directly involved in virulence control, it is an attractive target for anti-biofilm or anti-virulence compounds. However, screening of large libraries for inhibitors of diguanylate cyclases or phosphodiesterases is cumbersome due to the lack of simple and

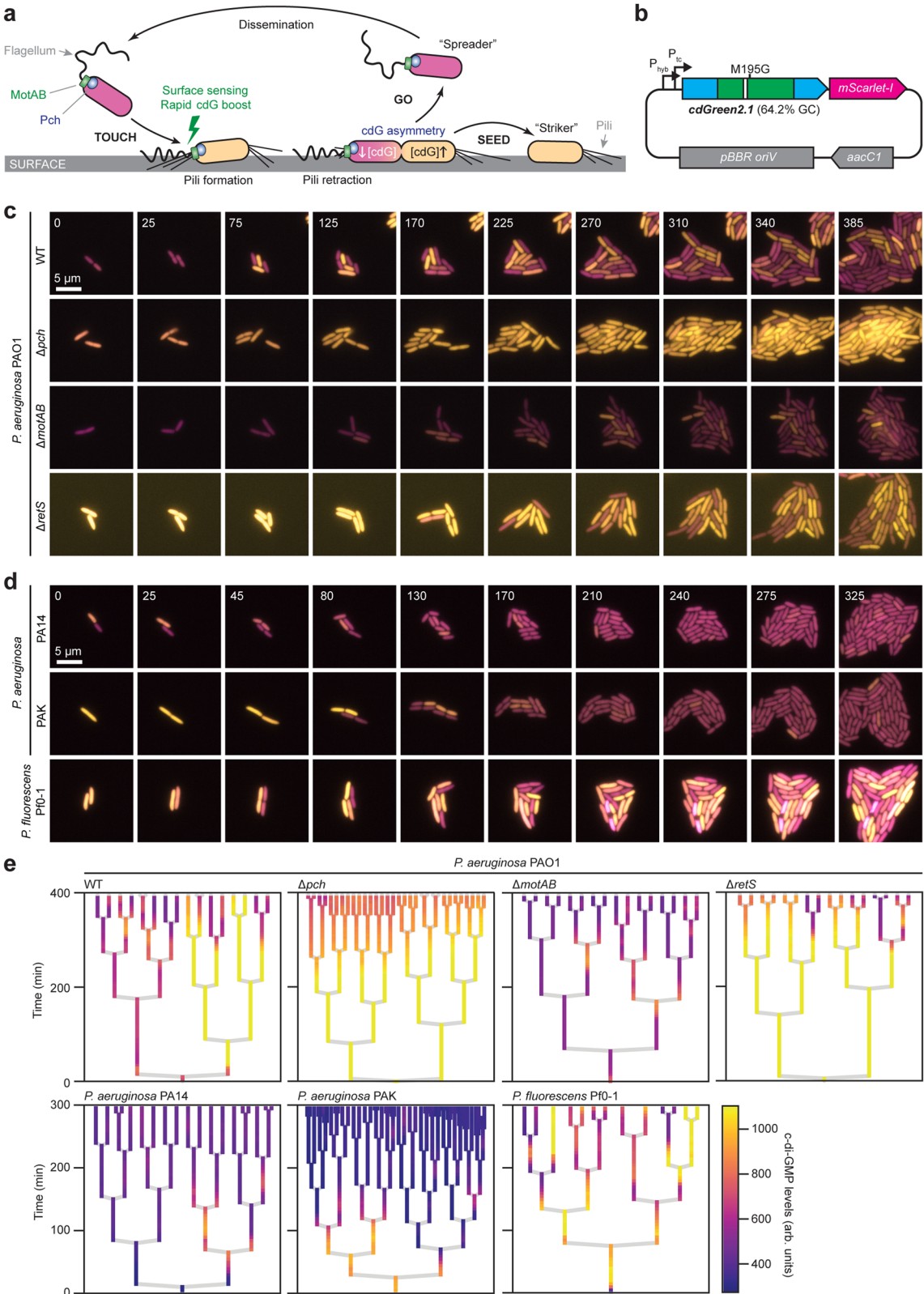

**Fig. 5 | C-di-GMP dynamics in surface-associated *Pseudomonas*. a** Schematic of the "touch-seed-and-go" model in *P. aeruginosa* PAO1. **b** Schematic of the plasmid used in *Pseudomonas* expressing cdGreen2.1 and the reference FP mScarlet-I. Note that cdGreen2.1, besides being high-GC-content optimized, also contains a single codon change (Met195Gly) that removes a potential internal start codon. $P_{hyb}$ and $P_{tc}$ represent two custom-designed nested, constitutive promoters. **c** Tracking of c-di-GMP levels in indicated *P. aeruginosa* PAO1 strains after spotting on agarose pads. Images are overlays of the cdGreen2.1 FITC/FITC signal pseudo-colored in yellow and the mScarlet-I signal pseudo-colored in magenta. Time stamps indicate minutes. Contrast and brightness settings are the same for all images. **d** Tracking of c-di-GMP levels in indicated *P. aeruginosa* strains after spotting on agarose pads. Images are overlays of the cdGreen2.1 FITC/FITC signal pseudo-colored in yellow and the mScarlet-I signal pseudo-colored in magenta. Time stamps indicate minutes. Contrast and brightness settings are the same for all images. **e** Lineage trees of indicated strains. arb. units, arbitrary units; cdG, c-di-GMP. Source data are provided as a Source Data file.

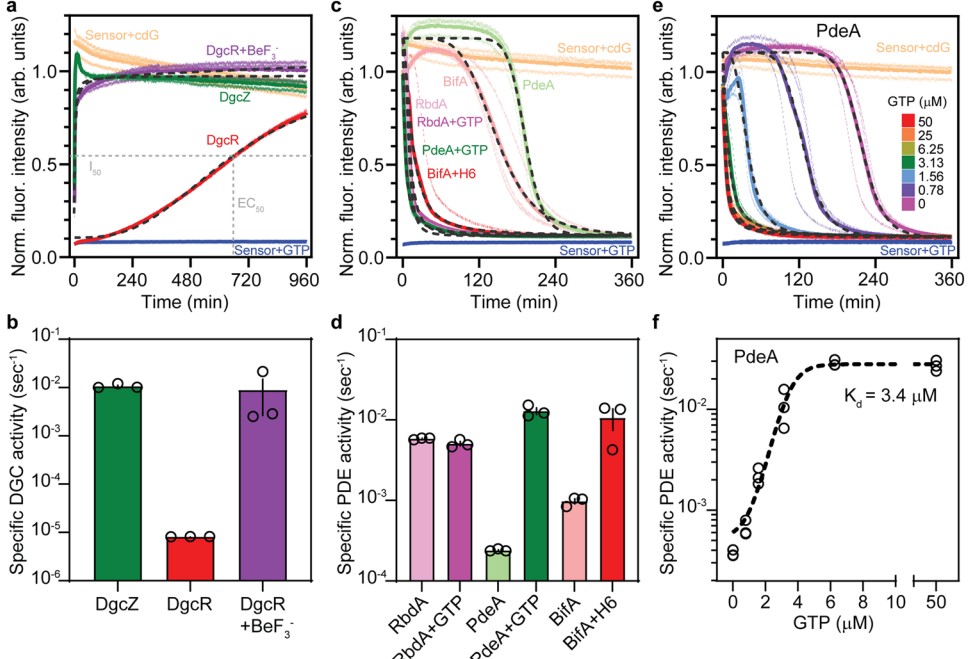

**Fig. 6 | In vitro diguanylate cyclase and phosphodiesterase activity assays using cdGreen2. a** C-di-GMP progression curves representing in vitro activities of selected diguanylate cyclases as measured with purified cdGreen2 (497 nm excitation, 530 nm emission). Baseline activities of cdGreen2 (sensor) with and without c-di-GMP (cdG) are indicated. Individual replicates ($n = 3$) (thin lines), the corresponding means (thick lines) and curve fits (dotted lines) are shown. $EC_{50}$ values were calculated from fitted curves and are defined as the time required to reach half-maximal intensity ($I_{50}$). The relationship between $I_{50}$ and $EC_{50}$ is illustrated for DgcR. **b** Quantification of specific activities of diguanylate cyclases from panel **a**. Shown are individual replicate values, means and SEMs ($n = 3$). **c** C-di-GMP progression curves representing in vitro activities of selected phosphodiesterase as measured with cdGreen2 (497 nm excitation, 530 nm emission). Baseline activities of cdGreen2 (sensor) with and without c-di-GMP are indicated for comparison. Individual replicates ($n = 3$) (thin lines), the corresponding means (thick lines) and curve fits (dotted lines) are shown. H6, H6-355-P1. **d** Quantification of specific phosphodiesterase activities using the fits from panel **c**. Shown are individual replicate values, means and SEMs ($n = 3$). **e** C-di-GMP progression curves representing in vitro activity of PdeA in the presence of different concentrations of GTP as measured with cdGreen2 (497 nm excitation, 530 nm emission). cdGreen sensor baselines with and without c-di-GMP are indicated for comparison. Individual replicates ($n = 3$) (thin lines), the corresponding means (thick lines) and curve fits (dotted lines) are shown. **f** PdeA-GTP dose-response curve as derived from the quantification of $EC_{50}$ values from data shown in panel **e**. Shown are individual replicates ($n = 3$) with the dotted line representing a fit used to calculate the $K_d$ (see Methods). arb. units, arbitrary units; cdG, c-di-GMP. Source data are provided as a Source Data file.

robust readouts. Given that cdGreen2 is able to accurately read out changes of c-di-GMP concentrations in real-time in vivo, we set out to adapt it to monitor c-di-GMP production and degradation in vitro. As a test case we purified the diguanylate cyclases DgcZ[67] from *E. coli* and DgcR[68] from *Leptospira biflexa*, as well as the phosphodiesterases PdeA[45,69] from *C. crescentus* and BifA[70,71] and RbdA[72] from *P. aeruginosa*. Purified cdGreen2 readily reported on the production of c-di-GMP by DgcZ and DgcR. In line with its stimulation by phosphorylation[68], DgcR activity increased dramatically in the presence of the phosphoryl-mimic beryllium fluoride (Fig. 6a, b). Similarly, purified cdGreen2 also accurately monitored the breakdown of c-di-GMP by all PDEs tested, recapitulating the known stimulatory effects of GTP on PdeA[69] and the anti-biofilm compound H6-335-P1 on BifA activity[71,73] (Fig. 6c, d). By curve fitting of c-di-GMP progression data, we next estimated $EC_{50}$ values to calculate the specific production and degradation rates of DGCs and PDEs from known initial substrate concentrations. Using a range of GTP concentrations, we were able to estimate the $K_d$ of GTP binding to PdeA to approximately 3.4 µM (95% CI: 3.1–3.9 µM), a value that closely matches previously reported data (4 µM) determined by an independent method[69] (Fig. 6e, f). Thus, our data demonstrate that cdGreen2 offers a convenient tool for the estimation of kinetic parameters for a range of DGCs and PDEs in vitro, although we appreciate that for PDEs pseudo-first-order conditions are not met. The possibility to monitor in vitro turnover of c-di-GMP in real time combined with the scalability of the setup make cdGreen2 a valuable tool for high-throughput screens for the discovery of anti-biofilm compounds that specifically target the activities of diguanylate cyclases or phosphodiesterases.

## Discussion

Understanding the complexity and dynamic behavior of biological systems requires methods to accurately and quantitatively monitor changes of small molecules and metabolites in real time and in individual cells. For example, the development of SFPB-based $Ca^{2+}$ indicators has allowed imaging of neuronal activity of individual cells in brains of live animals[74]; cAMP biosensors have spurred research in diverse areas of eukaryotic signal transduction[75-77], in particular those related to G-protein-coupled receptors and cyclic nucleotide-gated ion channels; and more recent $NAD^+$/NADH and NADPH sensors have uncovered the roles of these key metabolites in tumor progression and macrophage activation[78,79]. Biosensors also greatly facilitate drug discovery, as exemplified by the $NAD^+$/NADH biosensor SoNar, which has been employed in a high-throughput screen to isolate an anti-tumor agent that selectively induces cancer cell apoptosis[79].

Similar to eukaryotes, bacteria employ second messengers in a multitude of important signal transduction pathways, most notably the ubiquitous second messenger c-di-GMP. Although tools to monitor c-di-GMP in living bacteria have been developed, they are mostly based on species-specific transcriptional reporters or FRET probes and all lack one or more of the properties that define an ideal biosensor. This includes high selectivity and sensitivity, a dynamic range that spans the physiological window of the analyte and rapid and reversible ligand

binding. Also, the sensor should be (photo)stable enough to allow imaging over relevant biological time scales and the signal should be robust to noise, i.e., there should be ways to internally normalize the signal. Here, we developed two SFPBs, cdGreen and cdGreen2, with large dynamic ranges of up to 70-fold. While cdGreen is useful for robust steady-state measurements of c-di-GMP in the nanomolar range, its high affinity comes at the cost of slow dissociation, making it impossible to monitor dynamic changes of c-di-GMP on physiologically relevant time scales. To remedy this problem, we have developed cdGreen2, which shows rapid off kinetics while only marginally compromising on the dynamic range, making it an ideal biosensor to monitor c-di-GMP dynamics both in vivo and in vitro. Importantly, both biosensors offer ratiometric readouts, providing robustness by correcting for cell-to-cell variations of biosensor concentrations. Because the spectral properties of the sensors are compatible with most standard microscopy, flow cytometry and plate reader setups, other laboratories should be able to adapt them without the need for specialized equipment. While the adaptation of cdGreen2 as an in vitro tool should be straight forward, in vivo use of the biosensors might need species-specific optimization, such as adaptation of codon usage, the expression system or the use of reference FPs for normalization instead of the intrinsic ratiometric readout. While optimization remains an iterative approach, a number of computational and synthetic biology tools are available that might aid in the process. For example, synthesis of codon-optimized genes is offered by several commercial suppliers at affordable cost and fast turnaround times. A collection of minimal, constitutively active, $\sigma^{70}$-dependent promoters derived from the *E. coli* consensus is available, many of which are functional in distantly related bacteria[80,81]. Alternatively, strong species-specific housekeeping promoters might be used or constructed using low-key semi-synthetic approaches[82–84]. Finally, translation rates can readily and predictably be tuned using ribosome binding site (RBS) engineering[85–87] to yield high protein levels even when the genetic tools to achieve high gene transcription levels are limited.

To demonstrate the power and versatility of this latest generation of biosensors, we have dissected the c-di-GMP-mediated genetic program that drives the *C. crescentus* bimodal life cycle and have visualized the behavior of individual cells of *P. aeruginosa* during their short-term adaptation to surfaces. *C. crescentus* has long served as a model system for bacterial cell cycle control and it has been postulated for a number of years that c-di-GMP oscillations play an important role in cell cycle progression and morphogenesis[3]. However, existing tools were unable to visualize c-di-GMP cell cycle oscillations and the importance and epistatic interactions of the various regulatory proteins in this network and their contribution to c-di-GMP turnover had remained ill-defined. Using cdGreen2, we now robustly visualize c-di-GMP fluctuations during the *C. crescentus* cell cycle and demonstrate the central importance of individual components of the network in establishing c-di-GMP bimodality. Similarly, cdGreen2 visualized and corroborated an asymmetric program that was postulated to drive *P. aeruginosa* growth and dissemination on abiotic and biotic surfaces, paving the way to carefully dissecting the underlaying regulatory network responsible for this behavior. We envisage that cdGreen2 will be an invaluable tool to dissect cell-to-cell variations and heterogeneity in most bacteria making use of c-di-GMP. The biosensor could also be a useful tool to dissect more complex communities such as biofilms and to untangle how dynamic and stochastic changes in c-di-GMP correlate with distinct cellular identities in mixed bacterial populations[71].

So far, we have used cdGreen2 primarily to track dynamic changes of c-di-GMP over time. Thanks to its advantageous imaging properties, it may also be possible to visualize spatial gradients of the second messenger. Several studies have proposed that some diguanylate cyclases control c-di-GMP-dependent processes in a highly specific manner through a direct interaction with their cognate effector proteins and

without affecting global cellular levels of c-di-GMP[88–90]. To explain this, "local c-di-GMP signaling modules" were postulated, in which specific DGCs and PDEs assemble with their target structures in larger complexes and specifically and locally activate downstream components[3,10,11]. However, insulated subcellular c-di-GMP microenvironments have not, to the best of our knowledge, been visualized so far. As this will require highly sensitive and dynamic sensors, cdGreen2 or equivalent tools could serve as an entry point for the development of next-generation biosensors that are able to read out spatial gradients of small molecules in bacteria. The design principles and functional enrichment strategies developed in this study could serve as a blueprint for the development of robust genetic probes for quantitative, dynamic in vivo imaging of other signaling molecules, metabolites or antibiotics. Multiplexing of different biosensors could eventually provide the opportunity to study the interplay of different regulatory or metabolic networks and investigate multiple physiological parameters of individual cells of bacterial communities or during colonization of the host with unprecedented resolution and integrative power.

## Methods

### Strains, plasmids, oligonucleotides and synthetic DNA
Strains, plasmids, oligonucleotides and synthetic DNA used in this study are listed in the Supplementary Data 1 file.

### Media and growth conditions
*Escherichia coli* DH5α, DH10B, TOP10 or commercial electro-competent cells (MegaX DH10b T1R and ElectroMax DH5α, see below) were used for cloning and were routinely cultivated in LB-Miller at 37 °C. The *E. coli* MG1655 derivative AKS494 (see below) was used for all biosensor experiments. *Pseudomonas* strains were grown in LB-Lennox at 37 °C overnight for pre-cultures; cultures used for microscopy analysis, were diluted back 200-fold from overnight pre-cultures in LB-Lennox and grown at 30 °C into exponential phase. *Caulobacter crescentus* strains were grown in PYE (0.2% [w/v] bacto peptone, 0.1% [w/v] yeast extract, 0.8 mM MgSO$_4$, 0.5 mM CaCl$_2$) at 30 °C overnight for precultures; for microscopy analysis, cultures were diluted back 50-fold in the same medium and grown for 4–6 h into exponential phase at 30 °C. If not mentioned otherwise, PYE contained tetracycline and was supplemented with 100 µM cumate for biosensor expression. For microscopy time-lapse experiments, strains were grown on 1% agarose pads with appropriate medium at 30 °C. When appropriate, media were supplemented with antibiotics at the following concentrations (liquid/solid media for *C. crescentus*; liquid/solid media for *E. coli*; liquid/solid media for *P. aeruginosa*; in µg ml$^{-1}$): kanamycin (Km) (-/-; 50/50; -/-), oxytetracycline (Tc) (1/5; 12.5/12.5; -/-), chloramphenicol (-/-; 20/30; -/-), gentamycin (-/-; 20/20; 30/30), ampicillin (−/−; 100/100; -/-), carbenicillin (-/-; 100/100; -/-). The isopropy-β-D-thiogalactopyranoside (IPTG) stock was prepared in ddH$_2$O at a concentration of 1 M. The 4-isopropylbenzoic acid (cumate) stock (100 mM) was prepared in 100% ethanol and used at a final concentration of 100 µM. Anhydrotetracycline (aTc) stocks (1 mM) were prepared in DMSO and used at a final concentration of 200 nM.

### Strain construction
AB607 was generated by P1*vir*-mediated transduction of the kanamycin resistance-marked *yhjH* deletion from the respective Keio collection strain[91,92] in strain MG1655, followed by FLP-mediated excision of the resistance cassette using pCP20 as described previously[93]. In brief, transductants were transformed with pCP20, selected on LB plates containing ampicillin at 30 °C, restreaked on LB at 42 °C to cure from pCP20, and finally checked for loss of both kanamycin and ampicillin resistance markers by restreaking on appropriate selective media. AKS486 is a derivative of *E. coli* strain AB607 carrying plasmid pAR99. AKS494 was generated by P1-mediated transduction of a kanamycin

resistance-linked *pdeL'-'lacZ* translational fusion in AKS486 as described previously[37]. In brief, a lysate of strain AB2986 carrying the *pdeL'-'lacZ* fusion was prepared by 10-fold back dilution of an AB2986 overnight culture in 5 ml of LB supplemented with 10 mM CaCl₂, addition of 10 µl of P1*vir* starter lysate (prepared from an MG1655 wild type) and incubation with shaking at 37 °C until lysis was complete; the lysate was stored over chloroform at 4 °C. For transduction, 1 ml of an AKS486 overnight culture in LB was supplemented with 10 mM CaCl₂ and 100 µl P1*vir*(AB2986) lysate, incubated without shaking at 37 °C for 30 min, Na-citrate was added to a final concentration of 100 mM to stop phage adsorption, incubated for 1 h at 37 °C with shaking, and the bacterial suspension was plated on LB plates containing kanamycin and 40 mM Na-citrate, followed by incubation at 37 °C overnight. Colonies were purified by restreaking twice on the same medium and once on LB supplemented with kanamycin only. The *C. crescentus* Δ*pdeA* Δ*pleD::nptII* mutant was constructed by electroporation of plasmid pNPTStet-pleDnptII into UJ4454[45], selection on PYE/Tc/Km plates, growth of a single colony overnight in liquid PYE/Km and plating on PYE/Km plates supplemented with 0.3% sucrose. Loss of the plasmid backbone was verified by testing for tetracycline sensitivity. A ϕCR30 lysate prepared on this strain was used to move the marked *pleD* null allele into strain NA1000 *shkA^{D369N,DD} tacA^{DD}* (AKS217)[19] by generalized transduction.

## Plasmid construction

Polymerase chain reaction (PCR) was performed with Phusion DNA polymerase (NEB) in a total volume of 50 µl in GC buffer containing 10% (v/v) DMSO, 400 µM of dNTPs, 400 nM of each forward and reverse primer, 100–200 ng of DNA template and 0.4 µl of Phusion DNA polymerase. pAR81 was constructed by PCR-amplification of *yhjH* from an *E. coli* MG1655 colony with primers 7323/7324 and cloning in pNDM220[94] via *Bam*HI/*Xho*I. pAR94 was constructed by PCR-amplifying a *parMR*-containing fragment from pNDM220 using primers 6801/6803, a *cat*-containing fragment from pKD3[93] with primers 6802/6804, fusion of the two fragments by SOE-PCR using primers 6801/6804 and cloning in pNDM220 via *Psc*I/*Aat*II. Plasmid pAR99 was constructed by subcloning a fragment carrying *yhjH* from pAR81 into pAR94 via *Bam*HI/*Xho*I. pATTPtetycgR1 was constructed as follows. *ycgR* was PCR-amplified with primers 12056/12057 using an *E. coli* TOP10 colony as template. *Ptet-tetR* was PCR-amplified with primers 11469/12059 using plasmid pAR83 as template, followed by a second PCR with primers 12058/12059 using the first PCR product as template. pAR83 was a gift from Attila Becskei (Biozentrum, University of Basel) and encodes an autoregulatory P*tet*-TetR system. The two fragments, *ycgR* and *Ptet-tetR*, were joined by SOE-PCR, digested with *Pci*I/*Aat*II and cloned into pATT-Dest[29] digested with the same enzymes. pATT-Dest was a gift from David Savage (Addgene plasmid #79770). pATTPtetycgR2, which contains a unique *Eco*RI site, was constructed by PCR-amplification of a fragment from pATTPtetycgR1 using primers 12157/12059, digestion with *Pci*I/*Xba*I and cloning in the same sites of pATTPtetycgR1. pATTPtetycgR3, which contains a strong ribosome binding site upstream of *ycgR*, was constructed by PCR-amplification of a fragment from pATTPtetycgR2 using primers 12435/12436, digestion with *Afe*I/*Eco*RI and cloning in the same sites of pATTPtetycgR2. pBldDtemp was generated by PCR-amplification of *bldD* with primer pairs 14252/14253 and 14254/14255 from plasmid pIJ10663[30], fusion of the two fragments by SOE-PCR lacking primers and cloning in *Eco*RI/*Spe*I-digested pATTPtetycgR3 using Gibson assembly[95]. pBldDtemp contains a tandem fusion of *bldD* with *Bsa*I restriction sites in between *bldD* copies to allow domain/linker insertion using Golden Gate cloning[96]. p2H12 was isolated as a 4th-generation c-di-GMP sensor by FACS (see below). p2H12Matry was constructed by PCR-amplification of mScarlet-I from pBAD24_VCA0107_mScarlet-I_Cterm_04 with oligos 15688/15689 and cloning in *Kpn*I-digested p2H12 via Gibson assembly. p2H12Matry-blind was constructed in two

steps. First two fragments were PCR-amplified with primer pairs 15631/15633 and 15632/15634 from plasmid p2H12 and cloned in *Kpn*I/*Eco*RI-digested p2H12 using Gibson assembly. Next, the plasmid was digested with *Kpn*I and mScarlet-I was PCR-amplified from p2H12Matry with oligos 15688/15689 and assembled via Gibson cloning. p2H12ref was constructed by PCR-amplification of mScarlet-I from pBAD24_VCA0107_mScarlet-I_Cterm_04 with oligos 15549/15550 and cloning in *Spe*I-digested p2H12 via Gibson assembly. For construction of pConRef-2H12, oligos 16066/16067 were phosphorylated and annealed, and ligated with p2H12ref digested with *Xho*I/*Nde*I, replacing the original *Ptet* promoter and most of *tetR*. pConRef-2H12.D11 was constructed by subcloning a fragment encoding 2H12.D11 from pQFmcs-2H12.D11 (see below) into pConRef-2H12 via *Eco*RI/*Spe*I. pQFmcs was constructed by annealing phosphorylated oligos 14670 and 14671 and cloning the resulting dsDNA fragment in pQF[82] digested with *Eco*RI/*Spe*I. pQFmcs-2H12 was constructed by subcloning a fragment encoding 2H12 from p2H12 into pQFmcs via *Eco*RI/*Spe*I. pQFmcs-2H12.D11-scarREF was constructed by subcloning a *Eco*RI/*Nhe*I fragment from pConRef-2H12.D11 into pQFmcs-2H12.D11 digested with *Eco*RI/*Spe*I. pAK206-2H12.D11 was constructed by subcloning a *Eco*RI/*Spe*I-fragment encoding 2H12.D11 from pQFmcs-2H12.D11 into pAK206[97] digested with *Nhe*I/*Eco*RI. Final pAK206-2H12.D11 showed an unusual *Spe*I/*Nhe*I scar (ACTAGTAGC, instead of the expected ACTAGC), leaving the *Spe*I site intact. For construction of pBBR15-2H12.D11, oligos 16583/16584 were used for round-the-horn PCR with pAK206-2H12.D11 as a template. Plasmid pBBR15.2-2H12.D11opt was constructed in two steps; first, a *Spe*I on the backbone was eliminated by PCR-amplification of pBBR15-2H12.D11 with oligos 18655/18656 and circularization of the PCR product using Gibson assembly; next, a high-GC-content-optimized variant of 2H12.D11 encoded on two DNA fragments synthesized by Twist Bioscience (2H12.D11opt_frag1 and 2H12.D11opt_frag2; see Supplementary Data 1) was cloned in between *Nhe*I and *Spe*I sites of this construct via Gibson assembly. pBBR15.2-2H12.D11opt-scar was constructed by subcloning a *Spe*I/*Nhe*I-fragment encoding mScarlet-I from pConRef-2H12 in pBBR15.2-2H12.D11opt digested with *Spe*I and dephosphorylated. pET28-2H12, pET28-2H12-Matry and pET28-2H12.D11 were constructed by PCR-amplification of a fragment from p2H12, p2H12-Matry or p2H12.D11, respectively, using primer pair 15629/15630 and cloning in pET28a via *Nde*I/*Sac*I. pNPTStet-pleDnptII was constructed by PCR amplification of *nptII* conferring kanamycin resistance from pAK405[98] using primer pair 18357/18358, two fragments encoding *pleD* up- and downstream regions using primer pairs 18403/18404 and 18405/18406 from *C. crescentus* genomic DNA, respectively, and cloning of the three fragments in pNPTStet[19] digested with *Spe*I/*Sph*I using Gibson assembly. pET28a_HisTag-RbdA(374-818) was constructed by PCR-amplification of part of *rbdA* encoding residues 374-818 with oligos 14376/14377, digestion with *Not*I/*Eco*RI and cloning into pET28a digested with the same enzymes.

## Construction of biosensor libraries: cdGreen

For the initial library ("library 1"), cpEGFP was PCR-amplified from pTKEI-Tre-C04[29] (Addgene plasmid #79754) using standard PCR conditions with the reaction mix containing the following primers: 14292, 14293, 14294, 14295, 14296, 14297, 14298, 14299, 14300, 14301, 14302, 14303. The following cycling program was used: 98 °C for 2 min; 98 °C for 10 s, 55 °C for 10 s, and 72 °C for 30 s (30 cycles); 72 °C for 1 min; and hold at 10 °C. After amplification, 1 µl of *Dpn*I was added to the reaction and incubated at 37 °C for 1 h to digest the DNA template, followed by purification of the PCR product using a NucleoSpin Gel and PCR Clean-Up Kit (Macherey–Nagel) with elution in 15 µl NE elution buffer. The library was constructed using Golden Gate Assembly in a total volume of 50 µl in T4 DNA ligase buffer containing 250 ng of plasmid pBldDtemp, 1.75 µg of purified cpGFP PCR product, 1000 units T4 DNA ligase (NEB) and 30 units BsaI-HFv2 (NEB). The reaction was

incubated 90 min at 37 °C, 15 min at 55 °C and 20 min at 80 °C and purified using a NucleoSpin Gel and PCR Clean-Up Kit (Macherey–Nagel) with elution in twice 15 μl ddH$_2$O. The plasmid library was concentrated in a Eppendorf Concentrator plus (20 min, 30 °C, V-AQ) to approximately 5 μl and transformed into 100 μl of ElectroMax DH5α-E Competent Cells (Cat# 11319019, Thermo Fisher Scientific). After 30 min of recovery at 37 °C in 4 ml of SOC medium, cells were added to 400 ml LB containing 100 μg/ml carbenicillin. 10 μl were plated on LB plates containing 100 μg/ml ampicillin to calculate the library size and the remainder was incubated at 37 °C with shaking (180 rpm) overnight. The library was estimated to comprise $1.9 \times 10^6$ individual clones. Plasmid library DNA was purified from 10 ml of culture using a GenElute Plasmid Miniprep Kit (Sigma-Aldrich) with elution in twice 20 μl of ddH$_2$O, followed by concentration in an Eppendorf Concentrator plus (20 min, 30 °C, V-AQ). The library was transformed into strain AKS494 by electroporation using a BioRad GenPulser (1.75 kV, 25 μFD, 400 Ohms) and transformations (estimated to contain $> 2 \times 10^8$ clones) were outgrown in 400 ml LB containing 100 μg/ml carbenicillin, 6 μg/ml chloramphenicol and 200 nM aTc. Libraries for linker optimization (libraries 2, 3 and 4) were constructed similarly, except that different primer sets and different templates were used for PCR amplification of cpGFP, GGA was performed by incubation of the reactions overnight at 37 °C and MegaX DH10B T1R Electrocomp Cells (Cat# C640003, Thermo Fisher Scientific) were used for cloning. In each round of linker optimization, two amino acid changes (one in the N-terminal and one in the C-terminal linker each) were allowed in all possible combinations. The top-performing biosensors were selected for the next round of linker optimization, "optimized" amino acids were retained and other positions were randomized as described above. Library 2: oligos 14812, 14813, 14814, 14815, 14816, 14817, 14818, 14819, 14820, 14821. 14822, 14823, 14824 and 14825; pBldD2 as template. Library 3: oligos 14990, 14991, 14992, 14993, 14994, 14995, 14996, 14997, 14998, 14999, 15000 and 15001; pBldDX19-13 as template. Library 4: oligos 15202, 15206, 15207, 15208 and 15209; pBldD-A8 as template. Libraries 2-4 contained $>10^6$ individual clones and were propagated in ElectroMax DH5α-E Competent Cells (for EcoKI methylation) before being transformed into the final screening strain AKS494 as described above, outgrown overnight and used to inoculate cultures for FACS the next day.

## Fluorescence-activated cell sorting (FACS)

Overnight cultures were diluted 100-fold in 5 ml of LB supplemented with 100 μg/ml carbenicillin, 6 μg/ml chloramphenicol and 200 nM aTc with (1 mM) or without IPTG and grown for 2 h at 37 °C in a drum roller (125 rpm). 500 μl of culture were added to 5 ml of PBS and kept on ice. All sorts were performed using a BD FACSAria III cell sorter equipped with a 488-nm blue laser, a 495 nm LP mirror and a 514/30 nm BP filter for detection of GFP fluorescence at the FACS Core Facility, Biozentrum, University of Basel. Cells were sorted in "purity" mode in 5 ml of LB containing 100 μg/ml carbenicillin, 6 μg/ml chloramphenicol and 200 nM aTc. For initial naïve libraries, between $10^6$ and $10^7$ cells grown without IPTG showing a positive GFP signal were sorted and grown overnight at 37 °C in a roller drum (125 rpm). To set a threshold for GFP-positive cells, AKS494 harboring plasmid pBldDtemp was used as a negative control. After initial sorts, one "selection cycle" consisted of one round of FACS of cultures grown without IPTG (high c-di-GMP levels) and sorting for high fluorescence, followed by outgrowth of sorted cells, back-dilution, growth in medium with 1 mM IPTG (low c-di-GMP levels) and sorting for low fluorescence. The gates for high and low fluorescence were defined as containing the top or bottom, respectively, 1–5% of the main GFP distribution. GFP-negative subpopulations (likely representing cells that had lost the biosensor plasmid) were sorted in the "c-di-GMP low" step together with the lowest 1–5% of the main population. After 6–10 selection cycles single clones were isolated after an additional FACS round for "high c-di-

GMP" and individually tested for their response to the high and low c-di-GMP regimes as described above for libraries. For clones showing the desirable c-di-GMP-dependent response in GFP fluorescence intensity, plasmids were isolated using a GenElute Plasmid Miniprep Kit (Sigma-Aldrich) and sequenced using standard primers EGFP-C-for (GTCCTGCTGGAGTTCGTG) and EGFP-N-rev (GCTTGCCGTAGGTGGC ATC) at Microsynth (Balgach, Switzerland).

## Construction of biosensor libraries and FACS analysis: 2H12-Aff2 and 2H12-Aff8

A library of variants with partially randomized c-di-GMP binding motifs in the 5′-encoded bldD$_{CTD}$ portion of plasmid p2H12 were generated by PCR amplification of two fragments from p2H12 using primer pairs 15937/ 15939 and 15634/15938, joining of the two fragment via SOE-PCR using primers 15634 and 15939 and cloning in p2H12 digested with EcoRI/KpnI via Gibson assembly. The Gibson assembly reaction was purified using a NucleoSpin Gel and PCR Clean-Up Kit (Macherey–Nagel) with elution in twice 15 μl ddH$_2$O and the entire reaction was transformed into 100 μl of ElectroMax DH5α-E Competent Cells (Cat# 11319019, Thermo Fisher Scientific) as described above. The library comprised about 500'000 clones and was subsequently transformed into AKS494 as described above and subjected to FACS. Iterative FACS to enrich for functional sensors was performed as described above for cdGreen with the exception that the "c-di-GMP low" condition was defined by adding 20 μM IPTG instead of 1 mM IPTG. This IPTG regime was chosen to enrich for cdGreen variants that would "turn off" at intermediate c-di-GMP concentrations, i.e. at higher c-di-GMP concentrations compared to the original biosensor, with the expectation that such variants would display a lower K$_d$, possibly due to faster off kinetics. After 5 selection cycles, single colonies were isolated and tested individually. Two plasmids, p2H12-Aff2 and p2H12-Aff8, were isolated, sequenced and chosen for further studies.

## Combinatorial screening of c-di-GMP binding motif variants in *C. crescentus*: cdGreen2

Individual mutations in motif 1 (SQRGD) or motif 2 (RQDD) in the BldD c-di-GMP-binding site were then constructed in either the 5′ or 3′ bldD$_{CTD}$ fragment of cdGreen. For mutations in the 5′ fragment, SOE-PCRs were performed with flanking primers 15634 and 15939 and mutagenic primer pairs 16130/16131, 16132/16133, 16134/16135 or 16136/16137 and p2H12 as template and cloning in p2H12 via EcoRI/ KpnI and Gibson assembly. For mutations in the 3′ fragment, SOE-PCRs were performed with flanking primers 16138 and 16139 and mutagenic primer pairs 16130/16131, 16132/16133, 16134/16135 or 16136/16137 and p2H12 as template and cloning in p2H12 via KpnI/SpeI and Gibson assembly. To introduce both motif 1 and motif 2 mutation of 2H12-Aff2 or 2H12-Aff8 in the 3′ bldD$_{CTD}$, SOE-PCRs with 16138 and 16139 and mutagenic primer pairs 16188/16190 or 16189/16191, respectively, and p2H12 as template and cloning in p2H12 via KpnI/SpeI via Gibson assembly. Mutations in the 5′- and 3′-encoded bldD$_{CTD}$s were combined by subcloning 3′-encoded bldD$_{CTD}$ variants in p2H12 derivative encoding 5′-encoded bldD$_{CTD}$ variants via KpnI/SpeI. cdGreen variants were subcloned into pQFmcs-2H12 (replacing original cdGreen) via EcoRI/SpeI. Individual plasmids were then transformed in *C. crescentus* NA1000, resulting strains were grown overnight in 5 ml PYE supplemented with 1 μg ml$^{-1}$ oxytetracycline and 100 μM cumate, diluted 20-fold in the same medium and grown for another 4−6 hrs before being imaged on 1% agarose PYE pads by microscopy using a DeltaVision system. Images were visually inspected for late predivisional cells that showed asymmetric distributions of GFP fluorescence in the two future daughter cell compartments.

## Protein expression, purification and storage

cdGreen, cdGreen-Matry and cdGreen2 were overexpressed in *Escherichia coli* BL21(DE3). Cells were grown at 37 °C in terrific broth medium containing 50 μg ml$^{-1}$ kanamycin to an OD$_{600}$ of 0.8–1.0.

Protein expression was induced by 0.1 mM IPTG, then the temperature was reduced to 20 °C and cells were harvested after overnight cultivation. Cells were lysed with a sonicator in 50 mM Tris-HCl pH 8.0, 250 mM NaCl, 40 mM imidazole, and the soluble biosensor was purified by immobilized metal-affinity chromatography on a Ni Sepharose column (Cytiva) and subjected to size exclusion chromatography on a Superdex S200 column (Cytiva) in sensor buffer (25 mM Tris-HCl pH 7.5, 150 mM NaCl, 10 mM MgCl2, 10 mM KCl, 5 mM β-mercaptoethanol). Protein-containing fractions were pooled and concentrated in Amicon Ultra units (Merck). Protein yields were about 20 mg L$^{-1}$. Purified biosensors were stocked at −80 °C for long-term storage. Conveniently, purified biosensors are also very stable at 4 °C, such that we routinely store a working stock of cdGreen2 (at a concentration of ~850 μM) for in vitro PDE and DGC activity measurements at 4 °C for several months and up to 1 year, without or only very little loss in sensor activity/performance observed. PdeA was expressed similarly and cells were lysed in a micro-fluidizer in 2 x PBS containing 500 mM NaCl, 20 mM imidazole and 2 mM β-mercaptoethanol, followed by Ni-NTA metal-affinity purification and buffer exchange in GA buffer (20 mM Tris-HCl pH 8.0, 200 mM NaCl, 5 mM MgCl₂) using a PD-10 column. DgcR, DgcZ, BifA and RbdA were expressed and purified as described[68,71,72,99].

## Isothermal titration calorimetry (ITC)

ITC experiments were performed at 30 °C in sensor buffer, using a Microcal iTC200 instrument. The concentration of cdGreen in the calorimeter cell was 25 μM and the concentration of cyclic-di-GMP in the injector syringe was 1000 μM. 1 × 0.4 μl and 19 × 1.8 μl injections of cyclic-di-GMP were performed with stirring at 750 rpm, and the resulting heats of injection were monitored with an inter-injection spacing of 180 s, and an averaging time of 1 s. As a control or the heat of dilution and dissociation of cyclic-di-GMP, we performed an experiment that was identical in all respects apart from the omission of cdGreen from the solution in the calorimeter cell. The raw data were baseline-corrected, integrated, and analysed using the web version of the software Affinimeter[100]. To correct for the heat of dilution and dissociation of c-di-GMP, the integrated heats of injection for cyclic-di-GMP injected into buffer were subtracted from the integrated heats of injection for the injection of cyclic-di-GMP into cdGreen. The resulting heats of injection were fitted to the model described below, specified using the model-builder function of Affinimeter, with parameter errors determined automatically in the software by the Jackknife method.

## ITC: Model fitting

We expected four binding sites for cyclic-di-GMP. The general scheme for the binding equilibrium is therefore the stepwise association of four cyclic-di-GMP molecules (cdG) to the protein (P):

$$P \rightleftharpoons P \cdot cdG \rightleftharpoons P \cdot cdG_2 \rightleftharpoons P \cdot cdG_3 \rightleftharpoons P \cdot cdG_4$$

Macroscopic association constants and enthalpy change for binding can be assigned to each step ($K_1 - K_4$, $\Delta H_1 - \Delta H_4$, for each step left to right in the equilibrium above). The ITC titration of cyclic-di-GMP shows two clear phases, one at a 2:1 molar ratio of c-di-GMP and protein, and a second at a molar ratio of 4:1 of c-di-GMP and protein, with different enthalpy changes for binding evidenced by the different heat of injection in the initial plateau of each phase (ca −16 kcal/mol for the first phase, −7 kcal/mol for the second phase). This confirmed the anticipated total stoichiometry of 4:1 c-di-GMP to protein. The simplest equilibrium scheme that can fit such data comprises two sets of two equivalent binding sites. In terms of macroscopic association constants, this model fits $K_1$ and $K_3$, and sets $K_2 = 0.25^* K_1$ and $K_4 = 0.25^* K_3$ according to the expected statistical factor for two equivalent binding sites. The microscopic binding constants (site constants) for each set of equivalent sites can be obtained from $K_{micro1,2} = 0.5\ K_1$ and

$K_{micro3,4} = 0.5\ K_3$. In terms of enthalpy changes for binding, this model fits $\Delta H_1$ and $\Delta H_3$, and sets $\Delta H_2 = \Delta H_1$ and $\Delta H_4 = \Delta H_3$. The model is not a perfect fit to the data. It is probable that the underlying equilibrium is more complicated (e.g. having non-equivalent binding sites, and cooperativity) but inclusion of further fitting parameters without additional information or data would result in an over-parameterised fit. The fit gave $K_1 = 3.98 \times 10^8\,M^{-1}$ ($\pm 0.64 \times 10^8\,M^{-1}$), $K_3 = 6.05 \times 10^5\,M^{-1}$ ($\pm 1.18 \times 10^5\,M^{-1}$), $\Delta H_1 = -16.2$ kcal/mol ($\pm 0.1$ kcal/mol), and $\Delta H_3 = -7.5$ kcal/mol ($\pm 0.3$ kcal/mol). A concentration correction factor of 0.96 was fitted for the protein. These macroscopic association constants correspond to microscopic $K_D$ (site constant) values of 5.0 nM for the higher-affinity set of two sites, and 3.3 μM for the lower-affinity set of two sites.

## In vitro characterization of cdGreen biosensors

All measurements were taken in a SynergyH4 plate reader (BioTek) with protein and c-di-GMP in a total volume of 200 μl sensor buffer (25 mM Tris-HCl pH 7.5, 150 mM NaCl, 10 mM MgCl2, 10 mM KCl, 5 mM β-mercaptoethanol) using 96-well black/clear flat bottom plates (Costar) at 30 °C. Emission and excitation spectra of purified cdGreen, cdGreen-Matry and cdGreen2 shown in Figs. 2a and 3d were recorded at 30 °C at a protein concentration of 1 μM with excitation and emission wavelengths detailed in the figures or figure legends. Where indicated, c-di-GMP was added to a final concentration of 30 μM and reactions were equilibrated at 30 °C for 2 hrs before taking measurements. Spectral scans were at 1 nm resolution using the minimal bandwidth of 9 nm. For scans shown in Figs. 2b and 3e, and dose-response curves shown in Figs. 2c and 3f, protein concentration was 150 nM and c-di-GMP concentrations as indicated in figures or figure legends. Sensor protein and c-di-GMP were mixed and equilibrated for 7 hrs at 30 °C to reach binding equilibrium (see Supplementary Fig. 7) before measurements were taken. Multiple data points were determined for each c-di-GMP concentration by repeating measurements of the same well every minute over a 10 min period. Excitation and emission wavelengths are specified in the figures or figure legends (minimal bandwidth of 9 nm). $K_d$ values were fitted using GraphPad Prism 9 software and the "[Inhibitor] vs. response - Variable slope (four parameters)" model. Additional nucleotides were purchased from Jena Bioscience and tested at a final concentration of 10 μM with in the presence of 150 nM sensor protein and with (10 μM) or without c-di-GMP after equilibrating reactions for 5 hrs at 30 °C. For stoichiometric titrations experiments, sensor protein concentration was 25 μM and c-di-GMP was added from a 5 mM stock solution in sensor buffer to give the final concentrations indicated in the figures or figure legends. Sensor and c-di-GMP were mixed and equilibrated for 30 min at 30 °C before measurements were taken. For "dissociation by dilution" experiments to determine $k_{off}$ rate constants, 7.5 μM sensor protein was mixed with 200 nM (for cdGreen) or 300 nM (for cdGreen2) c-di-GMP and equilibrated for 7 hrs at 30 °C. Solutions were then diluted 50-fold in sensor buffer (4 μl in 200 μl sensor buffer; final c-di-GMP concentration of 4 nM for cdGreen and 6 nM for cdGreen2) and measurements were started immediately (ca. 10 s dead time). To record the baseline, cdGreen (7.5 μM) was incubated without c-di-GMP, followed by an identical dilution step and readings were subtracted from the response observed in the presence of c-di-GMP. This correction was necessary because cdGreen alone showed an initial increase of fluorescence intensity after dilution. This protein dilution effect was not observed for cdGreen2, so it was not necessary to include such a control for this biosensor. $k_{off}$ values were fitted using GraphPad Prism 9 software and the "Dissociation - One phase exponential decay" model. "Global estimation" of $k_{on}$, $k_{off}$ and $K_d$ was performed using association kinetics with three concentrations of the ligand (25 μM, 12.5 μM and 6.25 μM) and with sensor protein concentrations of 150 nM. Reactions were started by adding 5 μl of c-di-GMP from a 40-fold stock solution to 195 μl protein solution and measurements

were taken immediately (dead time of ca. 10 s). Association and dissociation rate constants and the equilibrium dissociation rate constant were fitted globally using the "Association kinetics (two ligand concentration)" model in GraphPad Prism 9.

## DGC and PDE in vitro activity

Experiments with purified DGCs and PDEs were carried out in a total of 100 µl GA buffer (20 mM Tris-HCl pH 8.0, 200 mM NaCl, 5 mM $MgCl_2$) containing 300 nM cdGreen2 and substrates, enzymes and ligands as detailed below in a SynergyH1 plate reader (BioTek) in 96-well clear flat bottom tissue culture plates (Falcon) at 30 °C. Substrate concentrations were 12 µM c-di-GMP (for PdeA and RbdA), 20 µM c-di-GMP (for BifA and PdeA-GTP dose-response curves) or 250 µM GTP (for DgcZ and DgcR). Substrates were added to cdGreen2 in GA buffer at least 1 h before the reaction was started and equilibrated at 30 °C. 20 µl of DGC or PDE stock solutions (see below for final concentrations) were mixed with 5 µl of GTP (varying concentrations; see figures and figure legends), 5 µl H6-335-P1 (10 mM in DMSO) or 5 µl BeF$_3$- (a mix of 10 mM NaF and 1 mM $BeCl_2$) stock solutions, or with 5 µl ddH$_2$O, and preincubated at room temperature for 30 min. Then reactions were started by adding 5 µl of enzyme/ligand mixes to pre-equilibrated GA buffer containing sensor protein and substrate (see above). Final protein concentrations were as follows: DgcZ, 3.44 µM; DgcR, 3.84 µM; RbdA, 3.44 µM; PdeA, 3.88 µM; BifA, 2.12 µM. After addition of the enzymes (ca. 1 min dead time), cdGreen2 fluorescence signals were recorded every minute for 16 h with the following settings: 497 nm excitation, 530 nm emission, optics: bottom, gain 90; 405 nm excitation, 530 nm emission, optics: bottom, gain 90. All plates included control wells with sensor alone, sensor with c-di-GMP, GTP or other relevant ligands. Note that c-di-GMP and GTP controls in panels a and c in Fig. 6 are shared since those experiments were carried out in parallel in the same 96-well plate. For normalization, recorded values were divided by the values from control wells containing sensor plus a saturating concentration of c-di-GMP, averaged over the 16-h time course of the experiment. All experiments were carried out in triplicates. Curve fittings and estimation of $EC_{50}$ values were performed on individual or triplicate replicates in GraphPad Prism 9 using the "[Inhibitor] vs. response -- Variable slope (four parameters)" model with Bottom and Top parameter constraints "shared value for all data sets" within one set of experiments. Specific DGC activities were calculated using the following formula:

$$\frac{K_d(cdGreen2)}{EC_{50} * [DGC]}$$

where $K_d$(cdGreen2) is the dissociation constant of cdGreen for c-di-GMP (1.24 µM), $EC_{50}$ is the time after which the cdGreen2 signal is half-maximal (in sec) and [DGC] is the DGC protein concentration (in µM). Specific PDE activities were calculated using the following formula:

$$\frac{[cdG] - K_d(cdGreen2)}{EC_{50} * [PDE]}$$

where [cdG] is the initial c-di-GMP substrate concentration (in µM), $K_d$(cdGreen2) is the dissociation constant of cdGreen for c-di-GMP (1.24 µM), $EC_{50}$ is the time after which the cdGreen2 signal is half-maximal (in sec) and [PDE] is the PDE protein concentration (in µM).

## Flow cytometry

Exponentially growing cultures were sampled in ddH$_2$O and analyzed on a BD LSR Fortessa using Diva software (BD Biosciences). After standard SSC-H/FSC-H and SSC-H/SSC-W gating for singlets, cells were gated for mScarlet-I-positive cells using a yellow-green 561 nm laser, a 600 nm long-pass mirror and a 610/20 nm band-pass filter, and cells were analyzed for their emission signals (long-pass mirror: 505 nm; band-pass

filter: 512/25 nm) following excitation with the violet 405 nm or the blue 488 nm laser. 50'000 events in the mScarlet-I-positive gate were collected for each sample. Data were analyzed using FlowJo v10.0.6 (FlowJo LLC). Ratios of emission signals upon excitation with 488 and 405 nm were calculated and plotted in FlowJo using the built-in "Derived parameters" tool. See Supplementary Fig. 8 for gating strategies.

## Microscopy

Images were acquired using a DeltaVision system with softWoRx 6.0 (GE Healthcare) on a Olympus IX71 microscope equipped with a pco.edge sCMOS camera and an UPlan FL N 100X/1.30 oil objective (Olympus). Signals for the c-di-GMP sensors were acquired using 390/18 nm (DAPI) and 475/28 nm (FITC) excitation filters and a 523/36 nm (FITC) emission filter. mScarlet-I was imaged using 575/25 nm excitation and 632/60 nm emission (A594) filters. For time lapse experiments, the following illumination settings were applied (Excitation/Emission, Exposure time, Power, Imaging interval) for experiments shown in Fig. 3c: FITC/FITC, 10 ms, 50%, 5 min; DAPI/FITC, 5 ms, 50%, 5 min. For Fig. 3h and Supplementary Movie 1: FITC/FITC, 100 ms, 2%, 20 s; A594/A594, 100 ms, 2%, 20 s. For Fig. 4: FITC/FITC, 15 ms, 50%, 5 min; A594/A594, 25 ms, 50%, 5 min. For Fig. 5: FITC/FITC, 150 ms, 10%, 5 min; DAPI/FITC, 100 ms, 5%, 5 min; A594/A594, 150 ms, 5%, 5 min. For Supplementary Fig. 4: FITC/FITC, 15 ms, 50%, 5 min; DAPI/FITC, 10 ms, 50%, 5 min. For Supplementary Fig. 5c: FITC/FITC, 10 ms, 50%, 5 min; DAPI/FITC, 5 ms, 50%, 5 min; A594/A594, 100 ms, 100%, 5 min. Data shown in Fig. 2e was acquired using NIS Elements on a Nikon Eclipse Ti2 inverted microscope equipped with a Hamamatsu ORCA-Flash4.0 V3 Digital CMOS camera and a 100X/1.45 Nikon Plan Apo Lambda 100x Oil Ph3 DM objective. Signals for the c-di-GMP sensors were acquired using 395/25 nm and 470/24 nm excitation filters and a 515/30 nm emission filter. mScarlet-I was imaged using 575/25 nm excitation and 632/30 nm emission filters. Cells were imaged within 15 min after spotting on agarose pads. Images were analyzed using the Fiji[101] plugin MicrobeJ[102].

## Time-lapse image analysis

Time-lapse movies were visually inspected using Fiji[101] and trimmed to exclude later frames were cells overlapped. Subsequently, cells were segmented and tracked using the DeLTA 2.0[103] deep-learning based pipeline. For *P. aeruginosa* we used the default pre-trained model for both segmentation and tracking. For *C. crescentus* we retrained the segmentation model using our own training data (the default tracking model was used). Training data was obtained by segmenting cells using the Ilastik pixel classification pipeline[104], followed by postprocessing using custom written Python code, and manually curation using Napari (https://zenodo.org/record/3555620). Data analysis was performed using custom written Python code (available here: https://github.com/simonvanvliet/cdg_dynamics_caulobacter). Visual inspection of the processed data showed that segmentation errors were extremely rare, but tracking errors occurred regularly, especially at later time points. We therefore implemented and automated a screening method to filter out erroneous cell tracks. Specifically, we only included pairs of sister cells that were tracked for at least 12 frames (1 h) for *C. crescentus* or 8 frames (40 min) for *P. aeruginosa*. We only consider frames for which data for both sister cells were available (*i.e.*, the track of the longest-lived sister was trimmed to the same length of the shortest-lived sister). Moreover, we excluded pairs were one or both cells showed unexpectedly large jumps in cell length between two frames (a decrease of >8% or an increase of >8% (*C. crescentus*) or 12% (*P. aeruginosa*)). Finally, we excluded cell pairs where the summed length of the sister cells at birth deviated strongly (a decrease of >6% or an increase of >20%) from the cell length of the mother cell just before division. For the remaining pairs of cells, the c-di-GMP level was estimated as the mean fluorescence intensity in the GFP/FITC channel. We then grouped sister cells into two classes: a c-di-GMP-high and a

c-di-GMP-low class based on the average c-di-GMP level over the full lifetime of the cells.

## Statistics and reproducibility
No statistical method was used to predetermine sample size. No data were excluded from the analyses. The experiments were not randomized. The Investigators were not blinded to allocation during experiments and outcome assessment.

## Reporting summary
Further information on research design is available in the Nature Portfolio Reporting Summary linked to this article.

## Data availability
Microscopy and flow cytometry raw data, and source data for lineage trees, are available from https://zenodo.org/records/10950147[105]. Supplementary Movie 1 is available from https://zenodo.org/records/10965397[106]. All other data generated in this study are provided in the Source Data file. Source data are provided with this paper.

## Code availability
The code generated for the analysis of c-di-GMP levels in time-lapse microscopy experiments is available from https://github.com/simonvanvliet/cdg_dynamics_caulobacter and https://zenodo.org/records/10900332[107].

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

## Acknowledgements

We thank Mark Buttner, David Savage, Attila Becskei, Johannes Schneider and Marek Basler for plasmids; Timothy Sharpe (Biophysics Core Facility, Biozentrum, University of Basel) for expertise support with ITC experiments and advice on stoichiometric titrations; Janine Bögli and Stella Stefanova (FACS Core Facility, Biozentrum, University of Basel) for their technical support with FACS; and Fabienne Hamburger for help with cloning and strain construction. This work was supported by the Swiss National Science Foundation grants 310030B_185372 and 310030_208107 to U.J.; and by NCCR AntiResist funded by Swiss National Science Foundation grant 51NF40_180541 to U.J. S.v.V. was funded by an Ambizione grant from the Swiss National Science Foundation (grant nr. PZ00P3_202186) and by the University of Basel.

## Author contributions

U.J. and A.Ka. conceived and designed experiments. A.Ka. performed experiments. A.Ka., S.v.V and U.J. analyzed data. R.P.J., R.D.T., I.S., A.R., A.Kl. and T.M. contributed materials. U.J. and A.Ka. wrote the draft with input from all authors.

## Competing interests

The authors declare no competing interests.
