## [Peer Review File · Nature Communications]

REVIEWERS' COMMENTS

Reviewer #1 (Remarks to the Author):

The methodological study by Kaczmarczyk et al describes the design and application of a novel BldD-based c-di-GMP biosensor. The authors use here a very clever approach based on directed evolution and FACS sorting to develop the cdGreen and cdGreen2 biosensors.

Currently, there are many different c-di-GMP biosensors available, either based on riboswitches as sensing unit or proteins (BldD, YcgR). However, among these, cdGreen2 is the only tool which allows detection of c-di-GMP changes with a high temporal resolution.

This manuscript impresses by the huge amount of work that is incorporated and by scientific accuracy used throughout the study. However, I am struggling with the fact that cdGreen2 is presented as a tool “superior” to all other available biosensors and “easy to use”. The truth is, that usage of such biosensors in individual models in most cases requires high investment of time and efforts to make it work. As the authors describe, in the case of *Pseudomonas* they faced problems with autofluorescence / codon usage / expression system, which they needed to adjust. I would appreciate if the authors highlight these limitations more clearly.

I guess the fact that cdGreen2 has a reduced affinity for c-di-GMP and therefore higher OFF rate means that in vivo expression levels of the biosensor must be quite high to detect clear signals. This would especially apply when the overall c-di-GMP levels are rather low and below the cdGreen2 K_d, like e.g. in *E. coli* (about 150 nM). Can the authors comment on that? How much of the biosensor (on protein level) do you need in vivo to detect clear signals? Some species are clearly limited here in terms of expression systems.

Maybe the author can consider to reduce the density of figure 1. Personally, I think fig. 1c and b are not needed as the structure of the BldD-c-di-GMP complex is published elsewhere.

Reviewer #2 (Remarks to the Author):

The manuscript by Kaczmarczyk and colleagues describes engineering of a versatile fluorescence-based sensor protein for the bacterial second messenger c-di-GMP, designated cdGreen2. It has high selectivity and physiologically relevant dynamic range. Importantly, it is distinct from the c-di-GMP sensors developed earlier by having a fast response (half-life ~1.4 min), which make it suitable for c-di-GMP measurements in real time. The study authors tested performance of the sensor in two different bacteria, showing that it adequately reports on the changes c-di-GMP levels during the development in *Caulobacter* and surface-attachment behavior in *Pseudomonas*. cdGreen2 can also measure c-di-GMP levels in vitro and may replace expensive antibodies and HPLC-MS measurements, thus democratizing the c-di-GMP research. The study is a major advance in tool development for the c-di-GMP signaling field which encompasses bacterial virulence, biofilm formation and cell development. The study is clever and involves enormous amount of meticulously performed experimentation. The authors present their findings clearly, describing their findings in great detail.

This is a rare instance where no constructive criticism seems warranted. I have just a couple of minuscule issues:

- (i) The use of the word "online" (line 327, 337) is ambiguous. Consider replacing it with "in real time".
- (ii) SBFP  SFPB (line 354). Is this abbreviation warranted at all (used only twice in the text)?

Reviewer #3 (Remarks to the Author):

Overall:

My expertise is in sensor construction, directed evolution and fluorescent biosensors, biophysics, so I haven't commented on the biology, which I assume other reviewers are better suited for. Overall, in terms of the sensor design - it is pretty novel and interesting and clearly effective. The directed evolution is well performed and yielded a nice sensor and all of the biophysical characterization is performed at a very high standard. I will be honest - this is much better than most papers I review. I can't really find anything to fault. Well done - it's comprehensive, careful, well performed, rigorous, and interesting. Maybe there are problems with the biology and microscopy that I can't see but the protein engineering and in vitro analysis to me seems essentially flawless.

REVIEWERS' COMMENTS

Reviewer #1 (Remarks to the Author):

The methodological study by Kaczmarczyk et al describes the design and application of a novel BldD-based c-di-GMP biosensor. The authors use here a very clever approach based on directed evolution and FACS sorting to develop the cdGreen and cdGreen2 biosensors.

Currently, there are many different c-di-GMP biosensors available, either based on riboswitches as sensing unit or proteins (BldD, YcgR). However, among these, cdGreen2 is the only tool which allows detection of c-di-GMP changes is a high temporal resolution.

This manuscript impresses by the huge amount of work that is incorporated and by scientific accuracy used throughout the study.

We thank the reviewer for this very positive feedback.

However, I am struggling with the fact that cdGreen2 is presented as a tool “superior” to all other available biosensors and “easy to use”. The truth is, that usage of such biosensors in individual models in most cases requires high investment of time and efforts to make it work. As the authors describe, in the case of *Pseudomonas* they faced problems with autofluorescence / codon usage / expression system, which they needed to adjust. I would appreciate if the authors highlight these limitations more clearly.

We now explicitly mention these limitations on page 14 of the revised manuscript and additionally discuss possible ways to overcome them:

“While the adaptation of cdGreen2 as an *in vitro* tool should be straight forward, *in vivo* use of the biosensors might need species-specific optimization, such as adaptation of codon usage, the expression system or the use of reference FPs for normalization instead of the intrinsic ratiometric readout. While optimization remains an iterative approach, a number of computational and synthetic biology tools are available that might aid in the process. For example, synthesis of codon-optimized genes is offered by several commercial suppliers at affordable cost and fast turnaround times. A collection of minimal, constitutively active, σ^{70} -dependent promoters derived from the *E. coli* consensus is available, many of which are functional in distantly related bacteria^{80,81}. Alternatively, strong species-specific housekeeping promoters might be used or constructed using low-key semi-synthetic approaches⁸²⁻⁸⁴. Finally, translation rates can readily and predictably be tuned using ribosome binding site (RBS) engineering⁸⁵⁻⁸⁷ to yield high protein levels even when the genetic tools to achieve high gene transcription levels are limited.”

I guess the fact that cdGreen2 has a reduced affinity for c-di-GMP and therefore higher OFF rate means that *in vivo* expression levels of the biosensor must be quite high to detect clear signals. This would especially apply when the overall c-di-GMP levels are rather low and below the cdGreen2 Kd, like e.g. in *E. coli* (about 150 nM). Can the authors comment on that? How much of the biosensor (on protein level) do you need *in vivo* to detect clear signals? Some species are clearly limited here in terms of expression systems.

We agree with the reviewer that a certain expression level should be achieved to observe robust signal distinct from background fluorescence. While we have never accurately determined the copy number of

c-di-GMP biosensors *in vivo*, the concentration of the biosensor needed to observe a good signal should be largely independent of its K_d for c-di-GMP. In theory, the response of the biosensor to c-di-GMP is expected to be maximal when the biosensor concentration is around the K_d for c-di-GMP, or, more precisely, at the upper end of the dynamic range of the biosensor-ligand dose-response curve. Whether or not the biosensor can report on c-di-GMP levels in a given species largely depends on whether the c-di-GMP levels *in vivo* match the dynamic range of the biosensor. Practically, we have indeed observed that the signal becomes very dim when chromosomally expressing cdGreen2.1 in *Pseudomonas aeruginosa* PAO1 from the same promoter used for plasmid-borne expression (see Fig. 5b), even though the biosensor still properly reports on changes of c-di-GMP. To enhance the cdGreen2.1 signal, we simply exchanged the original promoter with promoters from the Anderson collection (see ref. 80). Thus, a combination of the approaches outlined above should help to obtain constructs that yield sufficient biosensor levels to observe robust signals in a diverse range of bacteria.

Maybe the author can consider to reduce the density of figure 1. Personally, I think fig. 1c and b are not needed as the structure of the BldD-c-di-GMP complex is published elsewhere.

We prefer to keep panels b and c since not all readers will be familiar with the already published structure of the BldD-c-di-GMP complex and we believe that it helps the reader to better understand the biosensor design.

Reviewer #2 (Remarks to the Author):

The manuscript by Kaczmarczyk and colleagues describes engineering of a versatile fluorescence-based sensor protein for the bacterial second messenger c-di-GMP, designated cdGreen2. It has high selectivity and physiologically relevant dynamic range. Importantly, it is distinct from the c-di-GMP sensors developed earlier by having a fast response (half-life ~ 1.4 min), which make it suitable for c-di-GMP measurements in real time. The study authors tested performance of the sensor in two different bacteria, showing that it adequately reports on the changes c-di-GMP levels during the development in *Caulobacter* and surface-attachment behavior in *Pseudomonas*. cdGreen2 can also measure c-di-GMP levels *in vitro* and may replace expensive antibodies and HPLC-MS measurements, thus democratizing the c-di-GMP research. The study is a major advance in tool development for the c-di-GMP signaling field which encompasses bacterial virulence, biofilm formation and cell development. The study is clever and involves enormous amount of meticulously performed experimentation. The authors present their findings clearly, describing their findings in great detail.

We thank this reviewer for the very positive feedback.

This is a rare instance where no constructive criticism seems warranted. I have just a couple of minuscule issues:

(i) The use of the word "online" (line 327, 337) is ambiguous. Consider replacing it with "in real time".

Done.

(ii) SBFP  SFPB (line 354). Is this abbreviation warranted at all (used only twice in the text)?

Done.

Reviewer #3 (Remarks to the Author):

Overall:

My expertise is in sensor construction, directed evolution and fluorescent biosensors, biophysics, no I haven't commented on the biology, which I assume other reviewers are better suited for. Overall, in terms of the sensor design - it is pretty novel and interesting and clearly effective. The directed evolution is well performed and yielded a nice sensor and all of the biophysical characterization is performed at a very high standard. I will be honest - this is much better than most papers I review. I can't really find anything to fault. Well done - it's comprehensive, careful, well performed, rigorous, and interesting. Maybe there are problems with the biology and microscopy that I can't see but the protein engineering and in vitro analysis to me seems essentially flawless.

We thank this reviewer for the very positive feedback.